JCB Journal of Cell Biology

REPORT

# The postsynaptic density in excitatory synapses is composed of clustered, heterogeneous nanoblocks

Rong Sun[1,2], James P. Allen[2], Zhuqing Mao[1,2], Liana Wilson[1,2], Mariam Haider[1,3], Baris Alten[4], Zimeng Zhou[1,5], Xinyi Wang[1,6], and Qiangjun Zhou[1,2]

The nanoscale organization of proteins within synapses is critical for maintaining and regulating synaptic transmission and plasticity. Here, we used cryo-electron tomography (cryo-ET) to directly visualize the three-dimensional architecture and supramolecular organization of postsynaptic components in both synaptosomes and synapses from cultured neurons. Cryo-ET revealed that postsynaptic density (PSD) is composed of membrane-associated nanoblocks of various sizes. Subtomogram averaging from synaptosomes showed two types (type A and B) of postsynaptic receptor–like particles at resolutions of 24 and 26 Å, respectively. Furthermore, our analysis suggested that potential presynaptic release sites are closer to nanoblocks with type A/B receptor–like particles than to nanoblocks without type A/B receptor–like particles. The results of this study provide a more comprehensive understanding of synaptic ultrastructure and suggest that PSD is composed of clustering of various nanoblocks. These nanoblocks are heterogeneous in size, assembly, and distribution, which likely contribute to the dynamic nature of PSD in modulating synaptic strength.

## Introduction

Cellular function is driven by the nanoscale spatial organization of multi-protein assemblies and their interactions with other molecular components and organelles. One such example is on neuronal chemical synapses. Chemical synapses are highly complex, asymmetric intercellular junctions specialized for rapid information transfer from a presynaptic neuron to a postsynaptic cell. The structural diversity of synapses at different levels determines their functional diversity. For instance, the protein composition and subcellular structures of inhibitory and excitatory synapses differ significantly (Loh et al., 2016; Tao et al., 2018; van Oostrum et al., 2023), as do the protein composition and ultrastructure of synapses across different organs and brain regions (Harris and Weinberg, 2012; van Oostrum et al., 2023). Compelling evidence suggests that multiple presynaptic, postsynaptic, and cell adhesion proteins in excitatory synapses are finely organized in lipid-bound nanoscale structures to facilitate extremely rapid, dynamic, efficient, and tightly regulated transmission of information between neurons (Chamma et al., 2016; Glebov et al., 2017; Hruska et al., 2018; MacGillavry et al., 2013; Martinez-Sanchez et al., 2021; Maschi and Klyachko, 2017; Nair et al., 2013; Perez de Arce et al., 2015;

Tang et al., 2016). Moreover, the activity-dependent rearrangement of these synaptic proteins plays a key mechanistic role in synaptic plasticity, which is essential for learning and memory formation.

Recent studies provide evidence for a high-level organization of nanoscale supramolecular assemblies spanning the synaptic cleft, giving rise to nanocolumns of juxtaposed pre- and postsynaptic compartments in excitatory synapses (Biederer et al., 2017; Chen et al., 2018; Frank and Grant, 2017; Gramlich and Klyachko, 2019; Guzikowski and Kavalali, 2021; Kellermayer et al., 2018; MacGillavry et al., 2013; Tang et al., 2016). Specifically, presynaptic active zone scaffold proteins, such as Rab3 interacting molecule (RIM), form nanodomains that define synaptic vesicle docking and release sites. These sites align with clustered neurotransmitter receptors such as α-amino-3-hydroxy-5-methyl-4-isoxazolepropionic acid receptor (AMPAR) and postsynaptic scaffolding proteins such as PSD95 via adhesion proteins in the synaptic cleft (Haas et al., 2018; Martinez-Sanchez et al., 2021; Ramsey et al., 2021).

The postsynaptic density (PSD) appears in conventional electron micrographs as an electron-dense lamina just beneath

[1]Department of Cell and Developmental Biology, Center for Structural Biology, Vanderbilt Kennedy Center, Vanderbilt University, Nashville, TN, USA; [2]Vanderbilt Brain Institute, Vanderbilt University, Nashville, TN, USA; [3]Center for Structural Biology Cryo-EM Facility, Vanderbilt University, Nashville, TN, USA; [4]Department of Pharmacology, Vanderbilt University, Nashville, TN, USA; [5]School of Engineering, Vanderbilt University, Nashville, TN, USA; [6]Peabody College, Vanderbilt University, Nashville, TN, USA.

Correspondence to Qiangjun Zhou: qiangjun.zhou@vanderbilt.edu

B. Alten's current affiliation is Harvard Medical School, Boston, MA, USA and Department of Neurology, Massachusetts General Hospital and Brigham and Women's Hospital, Boston, MA, USA.



the postsynaptic membrane (Palay, 1956). A PSD meshwork model comprising regularly spaced vertical filaments was proposed based on EM and biochemical assays (Burette et al., 2012; Chen et al., 2008, 2011; DeGiorgis et al., 2006; Tao et al., 2018). Two additional forms of PSD organization have been proposed: nanodomains based on super-resolution imaging (Crosby et al., 2019; Hruska et al., 2022; MacGillavry et al., 2013; Nair et al., 2013; Pennacchietti et al., 2017; Tang et al., 2016) and liquid condensate based on *in vitro* PSD mixing assay (Feng et al., 2019; Zeng et al., 2016, 2018). All these studies provide more evidence for the high-level organization of nanoscale supramolecular assemblies in the synapse, indicating that synaptic proteins are not randomly distributed and instead form functional synaptic nanostructures. However, the precise nanoscale organization of the PSD and postsynaptic receptors is largely unknown due to the technical challenges imposed by synaptic molecular complexity.

Cryo-electron tomography (cryo-ET) is a powerful technique that produces 3D volume reconstructions, or tomograms, of macromolecular and subcellular biological structures in their native context at nanometer resolution. In this study, we first employed cryo-ET to image cultured hippocampal neurons, showing the intact synaptic structures at their near-native state. Then we developed a simplified sample preparation workflow to isolate synaptic terminals from mature rat hippocampi, achieving high-resolution synaptic nanostructures with cryo-ET and subtomogram averaging. Isolated synaptic terminals are commonly used to study synaptic function because they contain the complete presynaptic terminal, postsynaptic membrane, and the PSD (Evans, 2015). With this method, we directly visualized the plausible architecture of the transcellular organization in excitatory synapses. Our results revealed that PSD is formed by subsynaptic PSD nanoblocks of varying sizes. We also determined the structures of two types of postsynaptic receptor–like particles at 24-Å (type A) and 26-Å (type B) resolution. PSD nanoblocks exhibit a distinct spatial organization depending on whether they retain type A/B receptor–like particles. This finding underscores the intricate and diverse nature of PSD nanoblocks, further emphasizing the complexity of PSD nanostructures.

## Results and discussion

### Cryo-ET of intact synapses and isolated synaptic terminals reveals PSD subsynaptic organization

To study the subsynaptic organization of a synapse, we seeded dissociated primary hippocampal neurons on gold EM grids and examined the synaptic ultrastructure in intact neurons using cryo-ET (Fig. 1 A). In reconstructed tomograms, a typical synapse consists of a presynaptic compartment with a population of synaptic vesicles, a synaptic cleft, and a postsynaptic compartment with PSD (Fig. 1 A and Fig. S1, A–C, see also Video 1). Here, we focused on excitatory synapses that have thick PSDs of >30 nm (Palay, 1956; Tao et al., 2018) and excluded synapses with thin PSDs in cultured neurons which are likely inhibitory synapses from our analysis (Fig. S1, A–D).

A representative tomographic slice of a synapse (Fig. 1 B) reveals densities within the PSD. However, the crowded

molecular environment and the overall high thickness of intact neurons resulted in a low signal-to-noise ratio, making it challenging to identify and analyze protein complex patterns. To overcome these limitations, we utilized the software package IsoNet (Liu et al., 2022) to perform contrast transfer function (CTF) deconvolution and correct the missing wedge effect, enhancing contrast in tomograms of excitatory synapses. The application of IsoNet significantly improved structural interpretability (Fig. 1 C). After IsoNet correction, the PSD in Fig. 1 C clearly exhibited five density clusters with higher contrast compared with the PSD in Fig. 1 B. When we measured gray values of protein densities within a 40-nm-thick PSD region beneath the postsynaptic membrane (Fig. 1 C), quantitative analysis confirmed the presence of five distinct density clusters within the PSD (Fig. 1 D).

To further investigate subsynaptic organization at higher resolution, we optimized a streamlined workflow to isolate synaptic terminals from brain tissue for cryo-ET analysis (Fernández-Busnadiego et al., 2010; Martinez-Sanchez et al., 2021). Our preparations yielded two different types of isolated synaptic terminals based on their morphology: synaptoneurosomes, which retained an enclosed postsynaptic compartment (Fig. 1 E) (Hollingsworth et al., 1985; Quinlan et al., 1999), and synaptosomes, which retained a patch of postsynaptic membrane with PSD (Fig. 1 E, see also Video 2) (Evans, 2015; Gray and Whittaker, 1962). Their reduced sample thickness and lower molecular crowding provided higher-resolution structural information compared with intact synapses. This allowed us to directly visualize characteristic features of a synapse, including postsynaptic membrane particles, adhesion molecule–like particles, and synaptic vesicle–associated particles (Fig. S1, E–G). In our preparations, we successfully identified both synaptosomes and synaptoneurosomes, and no noticeable morphological differences were observed at nanoscale across different batches of sample preparation. All the synaptosomes we obtained were likely from excitatory synapses, as they have thick PSDs (Palay, 1956; Tao et al., 2018). In the representative synaptosome (Fig. 1, F and G), the PSD displayed seven distinct density clusters, a result further supported by gray value measurements of protein densities within a 40-nm-thick PSD region beneath the postsynaptic membrane (Fig. 1 H).

We observed that synaptosomes, synaptoneurosomes, and intact synapses exhibit distinct cleft widths (Fig. 1 I and Fig. S3 A), consistent with findings from previous cryo-ET studies (Held et al., 2024; Lucić et al., 2005; Tao et al., 2018). Specifically, synaptosome clefts tend to be wider than those of synaptoneurosomes and intact synapses (40.1 ± 4.4 nm for synaptosomes, $n = 74$; 32.0 ± 4.7 nm for synaptoneurosomes, $n = 41$, including all synaptoneurosomes from preparations with and without protease inhibitor and DTT; 30.8 ± 2.4 nm for intact synapses, $n = 34$). This result suggests that synaptosomes and synaptoneurosomes represent two relatively stable states after isolation. The disparity in cleft widths between synaptosomes and synaptoneurosomes may be due to differences in adhesion molecules that remain intact during the isolation process. Additionally, we segmented and measured synaptic vesicle diameters using CryoVesNet (Khosrozadeh et al., 2025). We observed

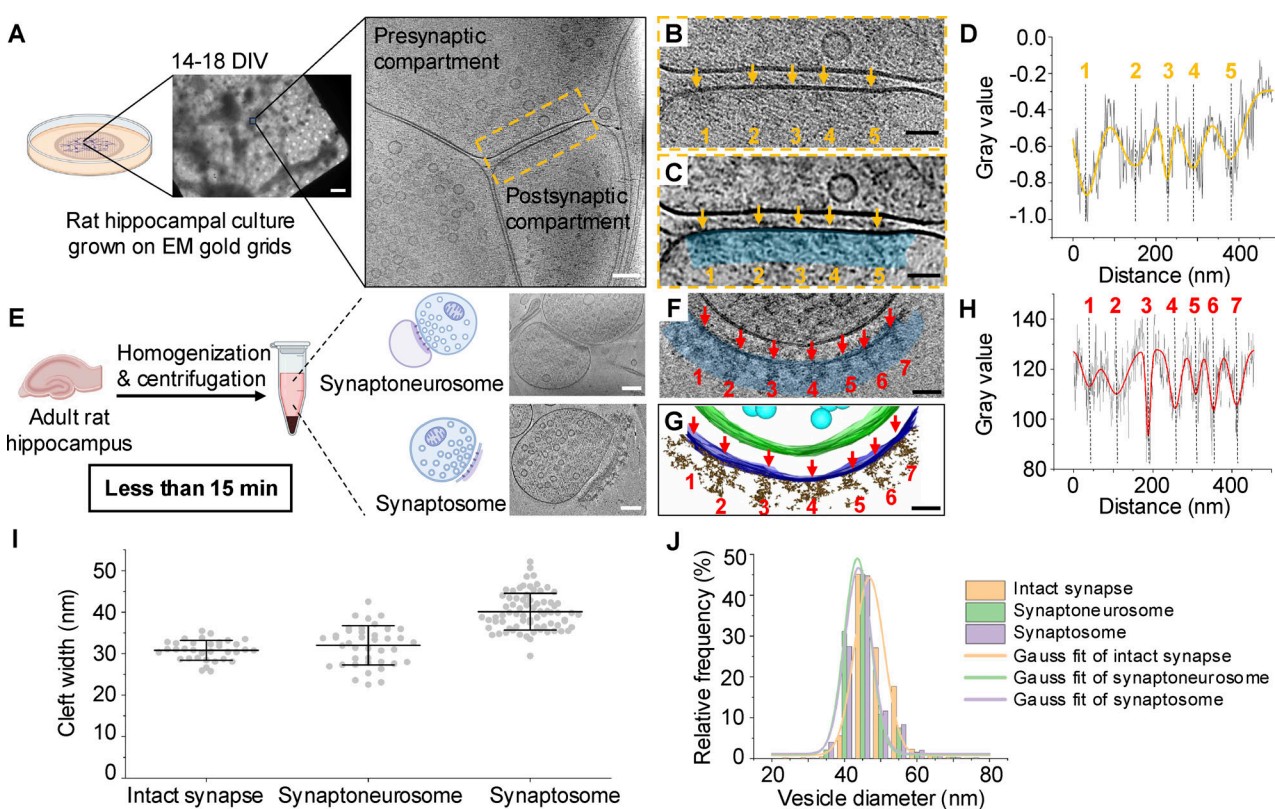

Figure 1. **PSD subsynaptic organization in intact neurons and isolated synaptic terminals. (A)** Workflow for imaging primary cultured neurons on gold EM grids using cryo-ET. **(B)** Zoom-in view of the synapse in A. **(C)** IsoNet-corrected tomographic slice of B. Orange arrows indicate PSD density clusters. Blue region refers to a 40-nm-thick region used in PSD density analysis. **(D)** Density plot shows average gray values (black lines) along the 40-nm-thick blue region in the panel of C. Orange lines indicate the fitting curve of average gray values. **(E)** Workflow for isolating and imaging synaptoneurosomes and synaptosomes using cryo-ET. **(F)** A tomographic slice showing an excitatory synaptosome without IsoNet correction. **(G)** 3D segmentation of the synaptosome shown in F. Red arrows indicate PSD density clusters. **(H)** Density plot shows average gray values (black lines) along the 40-nm-thick blue region in the panel of F. **(I)** Cleft widths of intact synapses (n = 34), synaptoneurosomes (n = 41), and synaptosomes (n = 74) with mean ± SD of each sample shown in the figure. **(J)** Synaptic vesicle diameter distribution of intact synapses, synaptoneurosomes, and synaptosomes. Scale bars: (A) middle panel, 5 μm; right panel, 100 nm; (B, C, F, and G) 50 nm; (E) upper right panel, 200 nm, bottom right panel, 100 nm.

that synaptic vesicle diameters were similar in both synaptosomes (43.40 ± 5.41 nm, n = 10,122 vesicles) and synaptoneurosomes (43.23 ± 5.16 nm, n = 6,153 vesicles), while vesicles in intact synapses were slightly larger (46.20 ± 4.55 nm, n = 11,270 vesicles) (Fig. 1 J and Fig. S3 B). Our results are consistent with previous cryo-ET studies of excitatory synapses that most synaptic vesicles are 40–50 nm (Fernández-Busnadiego et al., 2013; Fernández-Busnadiego et al., 2010; Held et al., 2024; Tao et al., 2018). These findings indicate that isolated synaptic terminal samples maintain consistent structural characteristics. In particular, the findings from both intact synapse and isolated synaptic terminal samples suggest that PSD subsynaptic organization is not homogeneous but instead composed of distinct density clusters.

## Quantitative analysis of PSD nanoscale organization

Visual inspection revealed three types of synaptosomes based on PSD organization: synaptosomes with separated PSD density clusters, synaptosomes with both separated and continuous PSD densities, and synaptosomes with continuous PSD densities (Fig. S1, H–J). Importantly, PSD density clusters were readily observable in the original 3D tomograms without requiring the

IsoNet correction applied to neuronal culture tomograms. To delineate the PSD nanostructures in 3D, we applied the following quantitative analysis. First, as shown in Fig. 1 F, we measured gray values of protein densities in a 40-nm-thick PSD region beneath the postsynaptic membrane. Then, we combined 2D gray values across consecutive virtual tomographic slices (Fig. 2 A) to generate a gray value density map (Fig. 2 B, top panel), which revealed the spatial distribution of projected gray values within the PSD. To analyze this distribution, we used density-based spatial clustering of applications with noise (DBSCAN) to quantitatively analyze the spatial distribution of gray values. DBSCAN is a density-based clustering nonparametric algorithm (Ester et al., 1996), which has been used in biological data analysis (Guzikowski and Kavalali, 2023; Rahbek-Clemmensen et al., 2017). The advantage of DBSCAN clustering is that it does not require predefining the number of clusters; instead, the algorithm identifies clusters based on the characteristics of the data. We identified distinct clusters with DBSCAN (Fig. 2 B, bottom panel) within the 40-nm-thick region. These clusters represent PSD densities flattened and projected onto the xz plane (Fig. 2 C). We also tested DBSCAN with different parameters (Fig. S2, C and D), and DBSCAN (6, 32) was applied for

Figure 2. **Quantitative analysis of PSD nanoblock. (A)** A schematic showing the measured 2D gray values in a 40-nm-thick PSD region beneath the postsynaptic membrane across consecutive virtual tomographic slices, which were combined to generate a gray value density map of the entire PSD region. **(B)** 3D density analysis of the synaptosome PSD region. Top panel: Density plot shows gray values of 40-nm-thick PSD density combined from multiple tomographic slices. Bottom panel: DBSCAN clustering result of 40-nm-thick PSD density projected onto the xz plane. Scale bars: 50 nm. **(C)** A schematic showing the clustering results of PSD densities produced from 3D synaptosome data. **(D)** Examples of PSD nanoblocks in 2D tomographic slices and corresponding projected DBSCAN clustering results. Values in each panel indicate the width (nm) of the PSD block. Blue dash lines in DBSCAN clusters indicate the corresponding planes of tomographic slices. Scale bars: 20 nm. **(E)** Distribution of PSD nanoblock area, width, centroid-to-centroid distance, and edge-to-edge distance of synaptosome, synaptoneurosome, and intact synapse.

synaptosomes isolated without protease inhibitor or DTT as the results were more consistent with the visualized electron density in the tomographic slices. Since it remains unclear whether these PSD protein clusters correspond to nanodomains, nanoclusters, or nanomodules previously described in fluorescent imaging studies (Broadhead et al., 2016; Hruska et al., 2018, 2022; MacGillavry et al., 2013; Nair et al., 2013; Tang et al., 2016), we refer to the PSD density clusters identified by cryo-ET (Fig. 2 B, bottom panel) as PSD nanoblocks.

To further quantitatively analyze each PSD nanoblock, we measured its area and width. The area was defined as its convex hull area, while the width of a nanoblock was defined as the maximum distance along the $x$ axis between two pixels within a cluster. We specifically chose to quantify in the $x$ axis to avoid the distortion from the missing wedge effect (Baumeister et al., 1999). Fig. 2 D shows examples of nanoblocks with different areas and widths. Area distribution analysis revealed multiple peaks across synaptosomes, synaptoneurosomes, and intact synapses, while width distribution was bimodal in synaptosomes and synaptoneurosomes and trimodal in intact synapses (Fig. 2 E). Furthermore, isolated synaptosome samples prepared without protease inhibitors or DTT from three independent experiments consistently yielded synaptosomes with PSD nanoblocks (Fig. S3, C and D). This consistency indicates that the clustering of PSD densities is a stable characteristic of the postsynaptic structure. We compared the distributions of nanoblock areas in synaptosomes, synaptoneurosomes, and intact synapses and found similar patterns, indicating that nanoblocks in all different samples have distinct sizes and tend to be some specific sizes. Our observations suggest that membrane-associated PSD components within synaptosomes are relatively stable and that they maintain structural integrity during the isolation process for synaptosome preparation via our simplified protocol. This finding is consistent with previous observations that PSD complexes and the PSD membranes are detergent resistant and stable enough to be purified from conventional synaptic fractions (Fernández et al., 2009; Frank et al., 2016; Frank et al., 2017; Jung et al., 2023). Thus, it is likely that the nanoscale organization and basic ultrastructure features we observed in synaptosomes and synaptoneurosomes also reflect synaptic architecture in vivo.

Peak values of the area fitting in Fig. 2 E suggested that large nanoblocks may be composed of several small nanoblocks, which is also supported by examining the morphology of some nanoblocks with large widths, as they can be divided into several small nanoblocks (Fig. 2 D). Median nanoblock areas were 543, 597, and 396 nm², and median widths were 30.4, 33.0, and 31.7 nm for synaptosomes, synaptoneurosomes, and intact synapses, respectively. While the median size of nanoblocks appears smaller than previously reported nanodomains observed via fluorescence imaging, larger nanoblocks (>50 nm widths along the x-axis, Fig. 2, D and E) are comparable in size to nanodomains, which range from 70 to 160 nm (Broadhead et al., 2016; Hruska et al., 2018; MacGillavry et al., 2013; Nair et al., 2013; Tang et al., 2016). This suggests that PSD nanoblocks might be a smaller subsynaptic composition of PSD compared with nanodomain. Moreover, it is possible that larger nanoblocks and

nanodomains are the same structure observed through different imaging techniques. Additionally, we measured the gaps between nanoblocks. The median centroid-to-centroid distances between nanoblocks were 37.1, 38.8, and 38.4 nm, while the nearest edge-to-edge distances were 6.6, 6.6, and 8.0 nm (Fig. 2 E). These gaps represent regions of lower protein density separating PSD nanoblocks. In addition to the aforementioned samples, synaptosomes isolated using isoosmotic homogenization buffer with protease inhibitor and DTT also show nanoblocks with similar patterns of dimensions (Fig. S3 E). Overall, our results consistently indicate that PSD protein distribution is not homogeneous. Instead, proteins from locally concentrated nanoblocks are segmented by less dense regions.

## Subtomogram averaging of postsynaptic receptor–like particles

To further explore the nanoscale organization of postsynaptic compartments, we employed the subtomogram averaging method to detect postsynaptic receptor-like particles in our synaptosome samples. AMPARs and n-methyl-d-aspartate receptors are prominent receptors in excitatory synapses and consist of N-terminal domains and ligand-binding domains within the synaptic cleft (García-Nafría et al., 2016; Greger et al., 2017; Zhao et al., 2019). Due to the relatively high resolution of tomograms of synaptosomes, we were able to directly visualize and pick receptor-like particles on the postsynaptic membrane (Fig. 3 A and Fig. S1 E).

We manually selected 1,565 protein particles on postsynaptic membrane from 28 synaptosomes with the sizes of extracellular part ranging from 10 to 14 nm for subtomogram averaging (Fig. 3 B). Eventually, we achieved an averaged type A structure at 24 Å with 391 particles (Fig. 3 C) and an averaged type B structure at 26 Å with 189 particles (Fig. 3 E). When we compare the geometric shape of the average structures with existing PDB models, type A and B particles seem to resemble the O-shaped and the Y-shaped AMPARs, respectively (Fig. 3, D and F).

## Type A particles show closer association with nanoblocks

With various types of receptor-like particles identified in the 3D tomograms of synaptosomes, we next superimposed these receptor-like particles and other synaptic elements onto the PSD DBSCAN clustering results (Fig. 4 A). To analyze the spatial distribution and alignment of receptor-like particles relative to PSD nanoblocks within synaptosomes, we then defined the distance from the receptor-like particle to the nanoblock centroid as $d1$ and the distance from the nanoblock centroid to the farthest nanoblock edge as $D$ (Fig. 4 B). To account for nanoblock size, we normalized distances as $d1/D$ to determine whether receptor-like particles fall within nanoblock range.

We then analyzed the relationship between receptor-like particles and PSD nanoblocks (Fig. 4, C and D). The distribution of $d1/D$ differed significantly between type A and B particles (two-sample Kolmogorov–Smirnov test, $n = 207$ for type A particles and $n = 93$ for type B particles, D = 0.2551, P = 3.54 × $10^{-4}$), with type A particles being closer to nanoblocks than type B particles (Fig. 4, C and D, Mann–Whitney test, $n = 207$, median

Figure 3. **Subtomogram averaging of receptor-like particles in synaptosomes. (A)** An example showing receptor-like particles on the postsynaptic membrane of a synaptosome. **(B)** Processing steps for subtomogram averaging. **(C–F)** Subtomogram averaged type A particle (C and D) and type B particle (E and F) with AMPAR model fitting. Models from PDB bank are fitted to the averaged electron densities. Yellow: PDB 5IDE. Red: PDB 6QKZ. Scale bars: (A) upper left panel, 100 nm; bottom right panel, 20 nm; (C–F) 5 nm.

= 0.71 for type A particles and $n$ = 93, median = 1.06 for type B particles, P = 7.44 × 10$^{-5}$). To compare the distribution of receptor-like particles with excluded particles during subtomogram averaging, we also examined undefined particles over the resolvable PSDs. We then combined type A and B particles as a single group, referred as type A and B particles. Though the $d1/D$ distribution of undefined particles was different from that of type A and B particles (Fig. 4, E and F, two-sample Kolmogorov–Smirnov test, $n$ = 457 for undefined particles and $n$ = 300 for type A and B particles, D = 0.13055, P = 3.53 × 10$^{-3}$), the median $d1/D$ values of undefined particles and type A and B particles were not significantly different (Mann–Whitney test, $n$ = 457, median = 0.84 for undefined particles and $n$ = 300, median = 0.91 for type A and B particles, P = 0.1041). In addition, 58.7% of type A and B particles were in the nanoblock range ($d1/D$ < 1), while 60.0% of undefined particles were in the nanoblock range ($d1/D$ < 1) (Fig. 4 F). These results suggest that exclusion of undefined particles during 3D classification was likely not because they were embedded more within PSD densities but rather due to conformational differences. Overall, type A particles are more closely associated with nanoblocks than type B particles, highlighting distinct organizational patterns within the synapse and suggesting that PSD nanoblocks may assemble with different receptor types.

**PSD nanoblocks do not exhibit a specialized alignment with potential release sites**

To investigate transsynaptic organization, we further analyzed the spatial distribution and alignment of docked, tethered, or partially fused vesicles—structures likely marking the locations of synaptic release sites (Fig. S1, A–C)—in relation to PSD nanoblocks in intact synapses (Fig. 5 A). Synaptic vesicles were projected onto the PSD DBSCAN clustering plane, and the distance from the projected vesicle centroid to nanoblock centroid was defined as $d2$ (Fig. 5 B). To account for nanoblock size, we normalized these distances as $d2/D$, where $D$ represents the distance from the nanoblock centroid to its farthest edge, to assess whether synaptic vesicles fell within nanoblock range.

We then analyzed the relationship between potential release sites and PSD nanoblocks. In tomograms of intact synapses and tomograms of synaptosomes, we compared actual vesicle-to-nanoblock distance ($d2/D$) with randomized vesicle-to-nanoblock distance. To pair each actual vesicle, the localization of a vesicle was randomized 100 times within the PSD bounds. In our randomization, each pixel in the PSD range had equal possibility of being assigned to the vesicle. We found that there was no significant difference between experimental and random $d2/D$ in either intact synapses or synaptosomes (Fig. 5, C and D), indicating that PSD nanoblocks do not show a specialized alignment with the potential release sites. This result is consistent with a recent study showing that scaffolding nanoclusters do not align with membrane-proximal synaptic vesicles (Held et al., 2024). These findings highlight the complexity of pre- and postsynaptic organization and underscore the need for further quantitative studies to elucidate the structural and functional relationships within transcellular synaptic assemblies.

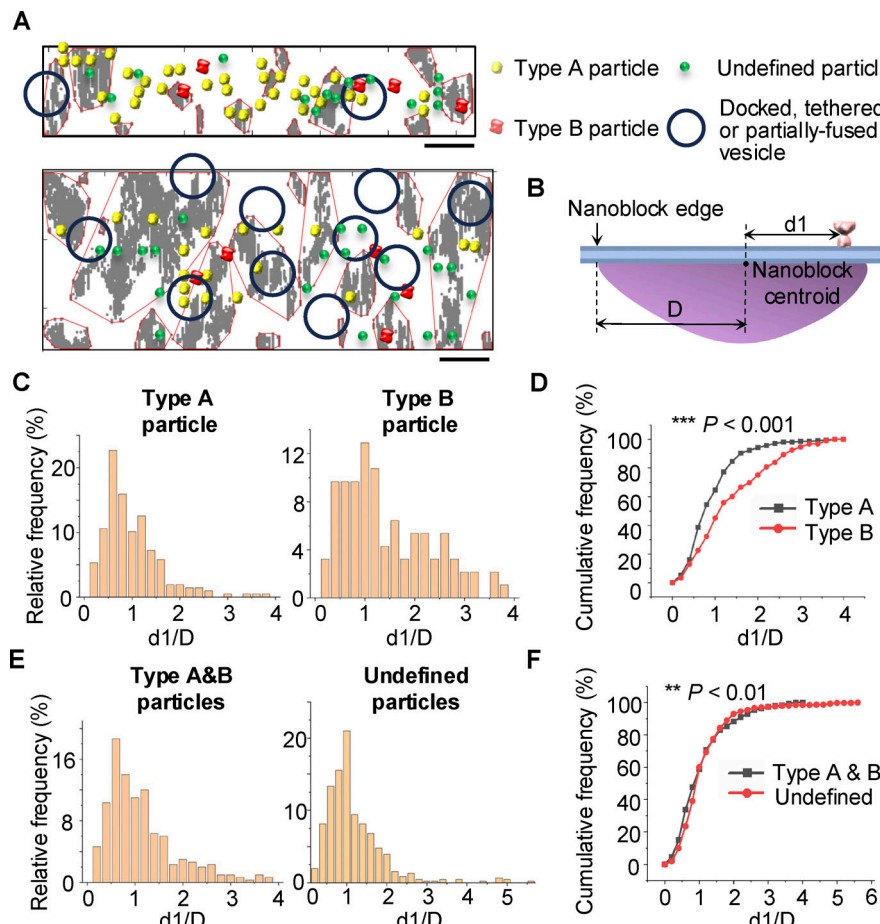

Figure 4. **Type A receptor–like particles are more closely associated with nanoblocks than type B particles. (A)** Examples show distribution of docked or tethered or partially fused vesicles, type A/B particles, undefined particles, and PSD nanoblocks projected onto the postsynaptic membrane in synaptosomes. Scale bars: 50 nm. **(B)** A schematic showing the distance $d1$ between receptor-like particle and the PSD nanoblock centroid and distance $D$ between the nanoblock centroid and the farthest edge. **(C and D)** Distribution of the normalized distance $d1/D$ between type A particles and nanoblocks and between type B particles and nanoblocks. Two-sample Kolmogorov–Smirnov test, $n = 207$ for type A particles and $n = 93$ for type B particles, D = 0.2551, P = $3.54 \times 10^{-4}$. **(E and F)** Distribution of the normalized distance $d1/D$ between type A and B particles and nanoblocks and between undefined particles and nanoblocks. Two-sample Kolmogorov–Smirnov test, $n = 300$ for type A and B particles and $n = 457$ for undefined particles, D = 0.13055, P = $3.53 \times 10^{-3}$.

### Nanoblocks containing type A/B particles align with potential release sites

Next, we examined the potential synaptic release sites and associated nanostructures in the cleft and in the postsynaptic compartment (Fig. 5 E). Surprisingly, we observed that tethered vesicles are in close proximity to the adhesion-like molecules, the membrane protein particles, and PSD densities (Fig. 5 E, top two panels and bottom left panel). This direct visualization of the pre-cleft-postsynaptic structure suggests that PSD scaffold proteins may link potential presynaptic release sites via adhesion molecules in the synaptic cleft. We also identified multiple cases where a PSD nanoblock aligned with adhesion molecules but lacked a tethered vesicle (Fig. 5 D, bottom right panel), indicating potential unoccupied release sites. We further analyzed the relationship between potential release sites and PSD nanoblocks in synaptosomes (Fig. 4 A).

Since PSD nanoblocks do not show specialized alignment with the potential release sites (Fig. 5, C and D), we analyzed three key components together: potential release sites, receptor-like particles, and PSD nanoblocks. Nanoblocks were classified into four types: those containing only type A particles, type B particles, both (mixed nanoblocks), or neither. In synaptosomes, we found that nanoblocks with type A/B particles had a significantly different distance ($d2/D$) from potential release sites compared with nanoblocks without any receptor-like particles (Fig. 5 F, Kruskal–Wallis ANOVA, $n = 20$ for type B nanoblock,

$n = 28$ for mixed nanoblock, $n = 119$ for type A nanoblock, and $n = 725$ for no A/B nanoblock.). This suggests that release sites are more closely aligned with nanoblocks containing type A/B particles. Furthermore, nanoblocks without receptor-like particles were smaller than those containing type A/B or type A particles (Fig. 5 G, Kruskal–Wallis ANOVA, $n = 5$ for type B nanoblock, $n = 4$ for mixed nanoblock, $n = 28$ for type A nanoblock, and $n = 147$ for no A/B nanoblock), suggesting larger nanoblocks are more active in recruiting receptor-like particles. These results indicate that release sites are positioned closer to nanoblocks containing type A/B particles, suggesting only certain types of PSD nanoblocks are aligned with potential presynaptic release sites for transsynaptic alignment and highlighting the functional heterogeneity of PSD nanoblocks. It also explains why there is no specialized alignment if we only consider the relationship between synaptic vesicles and PSD nanoblocks (Fig. 4, C and D).

The heterogeneity in the size, assembly, and spatial organization of PSD nanoblocks reveals further PSD complexity in response to activity-dependent molecular reorganization. Postsynaptic proteins situated between these nanoblocks may play a role in organizing smaller nanoblocks into larger, functional assemblies. Furthermore, the variation in nanoblock size and assembly likely reflects differences in protein composition, raising the possibility that nanoblocks of different sizes may have distinct molecular compositions and functional roles. It is likely the distribution of type A and B particles shown in Fig. 4, E

Figure 5. **Nanoblocks containing type A/B particles transsynaptically align with potential release sites. (A)** Examples show distribution of docked or tethered or partially fused vesicles and PSD nanoblocks projected onto the postsynaptic membrane in synapses. Circles indicate the positions of docked, tethered, or partially fused vesicles projected onto the postsynaptic membrane in intact synapses. **(B)** A schematic showing the distance $d2$ between synaptic vesicle (SV) and the PSD nanoblock centroid and distance $D$ between the nanoblock-centroid and the farthest edge. **(C)** Distribution of the normalized distance $d2/D$ between the experimental or randomized potential release site and nanoblock in intact synapses from cultured neurons. Two-sample Kolmogorov–Smirnov test, $n$ = 106 for experimental vesicles and $n$ = 10,600 for randomized vesicles, D = 0.09613, P = 0.28. **(D)** Distribution of the normalized distance $d2/D$ between the experimental or randomized potential release site and nanoblock in synaptosomes. Two-sample Kolmogorov–Smirnov test, $n$ = 145 for experimental vesicles and $n$ = 14,500 for randomized vesicles, D = 0.09448, P = 0.15. **(E)** Examples show adhesion molecule–like densities linking tethered synaptic vesicles to postsynaptic membrane particles and PSD nanoblocks on the postsynaptic membrane (top two panels and bottom left panel). Bottom right panel shows adhesion molecule–like densities aligning with postsynaptic membrane particles and PSD nanoblocks on the postsynaptic membrane, in the absence of a tethered synaptic vesicle. Red arrowheads indicate adhesion molecule–like densities in the cleft. **(F)** Normalized distance $d2/D$ between potential release site and centroid of different types of nanoblock. Median and 10–90% range are shown in the figure. Kruskal-Wallis ANOVA, $n$ = 20 for type B nanoblock, $n$ = 28 for mixed nanoblock, $n$ = 119 for type A nanoblock, and $n$ = 725 for no A/B nanoblock. P values are shown in the figure. **(G)** Area of different types of nanoblocks. Median and 10–90% range are shown in the figure. Kruskal-Wallis ANOVA, $n$ = 5 for type B nanoblock, $n$ = 4 for mixed nanoblock, $n$ = 28 for type A nanoblock, and $n$ = 147 for no A/B nanoblock. P values are shown in the figure. Scale bars: (A) 50 nm; (E) 20 nm.

and F reflected the distribution of different types of AMPARs in synapses. Our findings on receptor-like particle distribution align with studies showing that the nano-organization of AMPARs does not fully coincide with that of PSD95 (Hruska et al., 2018, 2022; Nair et al., 2013). This suggests that nanoblocks containing type A or B particles—or both—may correspond to nanoblocks containing AMPARs. Conversely, nanoblocks without detectable type A or B particles could represent spare scaffolds or nanoblocks containing other receptor types, such as n-methyl-d-aspartate receptors. Future investigations into the composition of PSD nanoblocks will be crucial for understanding the molecular organization and functions of the PSD.

A transsynaptic nanocolumn was discovered to align neurotransmitter release to receptors through super-resolution light microscopy (Tang et al., 2016). Thus far, only labeled synaptic proteins such as PSD95, receptors, and RIM have been shown to form nanoclusters as a part of nanocolumns. In contrast, cryo-ET allows us to examine the molecular organization of all cellular components, not only those that can be labeled for super-resolution light microscopy (Dubochet et al., 1988; Dubochet and Sartori Blanc, 2001; Taylor and Glaeser, 1974; Turk and Baumeister, 2020). Nanocolumns have been defined as transsynaptic structures that modulate evoked neurotransmitter release (Ramsey et al., 2021; Tang et al., 2016). Recent studies also suggest that spontaneous release and evoked release occur in

spatially distinct regions in the presynapse (Guzikowski and Kavalali, 2021; Wang et al., 2023). Along with our findings that there are different types of nanoblocks containing distinct receptor-like particles, these data raise the possibility that spontaneous release and evoked release may require their own dedicated nanoblocks. Moreover, nanoblocks without detected type A/B receptor–like particles may be spare scaffolds or contain other receptors. Future advances in cryo-ET may help to reveal a specific alignment for different subgroups of postsynaptic nanoblocks with different types of release sites by identifying each individual protein in the synapse.

## Materials and methods
### Synaptic terminal isolation
10-wk-old Sprague–Dawley rats of either sex were used for synaptosome isolation. The rats were kept in a 12 h:12 h dark:light cycle and provided with treats as well as cardboard enrichments before the experiments. The rats were deeply anesthetized with isoflurane. Hippocampi tissues were dissected in ice-cold isoosmotic homogenization buffer containing 0.32 M sucrose, 4 mM HEPES, and 1 mM EDTA, at pH 7.4 with or without protease inhibitor cocktail (1 tablet in 50 ml homogenization buffer) and 20 mM DTT. Then the tissue was homogenized at ~500 rpm for 4 strokes using an overhead stirrer (6480610; Electron Microscopy Sciences) and a tissue grinder (6479310; Electron Microscopy Sciences) in the ice-cold homogenization buffer. After homogenization, the homogenates were centrifuged at 800 $g$ at 4°C for 10 min. The supernatant was immediately taken for cryo-ET sample preparation. All animal procedures were performed in accordance with the guide for the care and use of laboratory animals and were approved by the Institutional Animal Care and Use Committee at Vanderbilt University.

### Primary dissociated neuronal cultures
The protocol was modified from previous works (Alten et al., 2021; Sun et al., 2019; Tao et al., 2018). Postnatal day 0 Sprague–Dawley rats of either sex were used for primary hippocampal cultures. Pregnant Sprague–Dawley rats were housed individually until they gave birth to a litter and were kept in a 12 h:12 h dark:light cycle. The pregnant rats were provided with the same treats as well as cardboard enrichments. Postnatal day 0 littermates were used to prepare primary dissociated neuronal cultures. Hippocampi were dissected in ice-cold 20% FBS containing Hanks' balanced salt solution. Tissues were then washed and treated with 10 mg/ml trypsin and 0.5 mg/ml DNase at 37°C for 10 min. The tissues were washed again, dissociated using a filtered P1000 tip, and centrifuged at 1,000 rpm for 10 min at 4°C. The pellet containing neurons was resuspended in Neurobasal Plus medium supplemented with GlutaMAX-I and B27 supplement. Neurons were plated onto Quantifoil R2/2 Au 200 EM grids coated with poly-L-lysine. Cultures were kept in humidified incubators at 37°C and gassed with 95% air and 5% $CO_2$. On Day 1 in vitro (DIV1), the Neurobasal Plus medium was replaced with 4 µM cytosine arabinoside–containing Neurobasal Plus medium. On DIV4, the cytosine arabinoside concentration

was dropped to 2 µM by performing a half media change. Cultures were kept without any disruption until DIV14. The cultures on EM grids were plunged frozen between DIV14–18, when synapses reached maturity. All animal procedures were performed in accordance with the guide for the care and use of laboratory animals and were approved by the Institutional Animal Care and Use Committee at Vanderbilt University.

### Isolated synaptic terminal and neuronal culture vitrification
#### Isolated synaptic terminal
FEI Vitrobot Mark III was used for plunge freezing. The parameters of the plunge freezer were as follows: humidity 100%, temperature 4°C, blot time 3.5 s, blot total 1, blot offset –1.5, and wait time 6 s. Supernatant after centrifugation mixed with 10-nm gold beads solution (cat # 25486; Aurion) at a 1:1 ratio was added to the mounted grid (R2/2 Cu 200 EM; Quantifoil). No other chemicals or stimulation was applied prior to plunge freezing. The grids were plunged into liquid nitrogen–cooled liquid ethane for rapid vitrification and were stored in liquid nitrogen until use.

#### Primary dissociated neuronal cultures
The vitrification process of neuronal cultures is similar to the synaptosome with a few modifications. The culture dishes were taken out from the incubator on DIV 14–18. The culture medium was replaced with a modified Tyrode's solution containing the following: (in mM): 150 NaCl, 4 KCl, 1.25 $MgCl_2$, 2 $CaCl_2$, 10 D-glucose, and 10 HEPES at pH 7.4. Then the cultured neurons were kept in the modified Tyrode's solution for 10 min. After setting up the plunge freezer as follows: humidity 100%, temperature 22°C, blot time 3.5 s, blot total 1, blot offset –1.5, and wait time 6 s, the grids with cultures were mounted. 4 µl 10-nm gold beads solution (in the modified Tyrode's solution) was added to the mounted grid before plunging. No other chemicals or stimulation was applied prior to plunge freezing. After rapid vitrification, the grids were stored in liquid nitrogen until use.

### Cryo-ET data collection
Automated batch tilt series collection was done with Thermo Fisher Scientific Tomography 5 software on Thermo Fisher Scientific Titan Krios G4 using the Gatan K3 camera in zero-loss mode (slit width 20 eV). The tilt series of neuronal cultures were collected using the dose-symmetric scheme (Hagen et al., 2017) starting from 0° to ±60° with an interval of 2° and with the defocus value at –7 to –10 µm, with the total electron dosage of ~150 e⁻/Å² that were evenly distributed between tilts. The final pixel size was 3.3 Å with 26,000× magnification. The tilt series of isolated synaptic terminals were collected with a voltage phase plate. The phase plate dataset was collected at the pixel size 3.3 Å with 26,000× magnification, using the dose-symmetric scheme starting from 0° to ±60° with an interval of 2° and with the defocus value at –1 to –2 µm. The total electron dosage of ~150 e⁻/Å² was evenly distributed between tilts. The phase plate was switched to a new position and activated to gain a phase shift of around 0.3π prior to each tilt series. Conditioning of the phase plate was also performed between each tilt. Before data collection on Krios, Thermo Fisher Scientific Glacios

was used for grid screening. In total, we acquired 123 tomograms of isolated synaptic terminals from three preparations in iso-osmotic homogenization buffer without protease inhibitor and DTT (99 synaptosomes and 24 synaptoneurosomes). We collected 211 tilt series of synapse-like structures from the wild-type primary hippocampal neurons and reconstructed them into 3D tomograms. Of these, 131 tomograms contained synapses, and 121 of these synapses exhibited thick PSDs.

### Data reconstruction and subtomogram averaging
#### Isolated synaptic terminal
Tomograms of synaptosomes at a pixel size of 3.3 Å with a phase plate were used for subtomogram averaging. Synaptosomes thicker than 500 nm were excluded, and 28 tomograms were selected. TOMOMAN package (Khavnekar et al., 2024; Wan, 2020) was used to preprocess the data. The workflow of TOMOMAN included motion correction by MotionCorr2 (Zheng et al., 2017), image sorting, dose filtering, and CTF estimation by Gctf (Zhang, 2016). Then the tilt series were aligned and reconstructed using IMOD (Kremer et al., 1996). Fine alignment of the tilt series was performed by using the 10-nm gold beads (cat # 25486; Aurion) as fiducial markers. Both back projection and back projection with five-iteration simultaneous iterative reconstruction technique (SIRT) were performed to reconstruct each tilt series. Segmentation and 3D rendering were done using IMOD. In the synaptosome dataset, 1,565 particles were picked from 28 synaptosomes according to the size and shape. Only the particles around 10–14 nm on the postsynaptic membrane were included. Subtomogram averaging was done using RELION 3.0.8 (Bharat and Scheres, 2016). We applied a box size of 96 using unbinned data. First, we generated the initial model using 3D auto-refine with the unbinned data without applying symmetry. 3D classification resulted in two good classes. After 3D classification, we generated 3D initial models for each class, respectively, with C2 symmetry and did 3D auto-refine for each class using the corresponding initial models. Manual repicking of the particles was performed to exclude the particles that were too close (distance <8 nm), and final averaging was performed using the remaining particles in 3D auto-refine.

#### Synapse
Like synaptosome and synaptoneurosome datasets, TOMOMAN package was used to preprocess the data. However, since the synapse dataset was not used for subtomogram averaging to achieve high resolution, CTF correction was not applied to the reconstruction. Fine alignment of the tilt series was performed by using the 10-nm gold beads and only five-iteration SIRT was performed to reconstruct each tilt series in IMOD. Segmentation and 3D rendering were done using IMOD. IsoNet (Liu et al., 2022) correction was performed using binning 4 five-iteration SIRT reconstructed tomograms. CTF deconvolve was done using SnrFalloff 1.0 and DeconvStrength 1.0. Refinement was performed using CTF-deconvolved tomograms with nose mode as noFilter. We have tested different iteration numbers in refinement and chose 30 iterations with a good missing wedge correction effect and no obvious artifact (Fig. S2, A and B).

Predicted tomograms were generated using model_iter30.h5 as the trained model.

### PSD density analysis and quantification
Density of PSD was measured with binning 4 tomograms using the following method. In the isolated synaptic terminal dataset, synaptosomes with a PSD or the postsynaptic compartment of synaptoneurosomes too close to the cryo grid carbon hole edge were excluded as the gray values were affected by the hole edge. Synaptosomes and synaptoneurosomes thicker than 500 nm and synaptosomes with the postsynaptic membrane not parallel to its presynaptic membrane were also excluded. 140 tomograms (58 tomograms for synaptosomes isolated without protease inhibitor or DTT in the homogenization buffer, 41 tomograms for synaptosomes isolated with protease inhibitor and DTT in the homogenization buffer, and 41 tomograms for synaptoneurosomes isolated with or without protease inhibitor and DTT in the homogenization buffer) were selected for PSD density clustering analysis. In the culture neuron dataset, synapses thicker than 700 nm were excluded, and 34 tomograms were selected for PSD density analysis. The gray values within 40-nm range from the postsynaptic membrane were projected to xz plane using Fiji (Schindelin et al., 2012). The gray value of each pixel of the projected 2D plane was the average value of the 40-nm column. Next, we used different strategies to process the dataset. For isolated synaptic terminal data, we measured the average gray value in empty space as background gray value. Then we chose the bottom 25% gray values (25% darkest) below the background value for the following clustering analysis for synaptosomes isolated without protease inhibitor or DTT in the homogenization buffer. And we chose the bottom 20% gray values (20% darkest) below the background value for the following clustering analysis for all synaptoneurosomes and synaptosomes isolated with protease inhibitor and DTT in the homogenization buffer. For synapses from cultured neurons, we analyzed the density in the IsoNet-corrected tomograms. The data set was normalized after IsoNet internal process, and we chose the bottom 20% gray values (20% darkest) for the following clustering analysis. Lastly, clustering analysis algorithm DBSCAN in MATLAB 2023b was adopted to do the clustering of PSD density. In the synaptosome sample analysis without protease inhibitor or DTT, DBSCAN with different parameters was tested (Fig. S1 C), and DBSCAN (eps = 6, minimum number of points = 32, eps represents the maximym distance between two samples for one to be considered in the neighborhood) was applied. In the synaptosome isolated with protease inhibitor and DTT, synaptoneurosome and synapse analysis, DBSCAN (eps = 7, minimum number of points = 32) was applied. The parameters were determined for different samples to be consistent with our visual inspections.

The area of a nanoblock was defined as its convex hull area shown by red lines in Fig. 2 B, bottom panel and Fig. 2 D, bottom panel. All area distributions were cut off to 3,500 nm² to show the fitting peaks. The width of a nanoblock was defined as the largest x-axis distance between two pixels in the cluster. The widths of nanoblocks are also marked in the bottom panel of Fig. 2 D. To precisely measure the height of a nanoblock, we first

averaged the tomographic slices in the range that contains the nanoblock determined in the DBSCAN clustering results. Then we drew a line to measure the gray value of the nanoblock on y axis at its largest length. The height of the nanoblock was defined as the distance from the farthest pixel of the nanoblock to the postsynaptic membrane. Since we took the weight of each pixel in the nanoblock as equal, the centroid of a nanoblock is the geometric centroid of the nanoblock by averaging all coordinates of the pixels of a nanoblock.

### Alignment analysis of different synaptic components

Positions of all tethered, docked, or partially fused vesicles and subtomogram averaged receptor-like particles were projected to the xz plane in the same coordinate system of PSD density to analyze the relation among potential release sites, receptor-like particles, and PSD nanoblocks for both synaptosomes (vesicle number is 106) and synapses (vesicle number is 145). $d1/D$ and $d2/D$ were defined as shown in Fig. 4 B and Fig. 5 B. Here, $d1$ was defined as the distance from a receptor-like particle centroid to a nanoblock centroid. $d2$ was defined as the distance from a projected synaptic vesicle centroid to a nanoblock centroid. $D$ was defined as the distance from the farthest pixel of the nanoblock to the centroid of the nanoblock. Thus, $d2/D$ is the normalized distance from a receptor-like particle to a nanoblock, and $d1/D$ is the normalized distance from a synaptic vesicle to a nanoblock. For the analysis of the relationship between receptor-like particles and PSD nanoblocks, it was conducted in the 16 synaptosomes, where PSD were resolvable and subtomogram averaging identified receptor-like particles exist. For the analysis of the relationship between potential release sites and PSD nanoblocks, it was conducted in the 32 tomograms of synapses and the 32 tomograms of synaptosomes, where potential release sites and PSD nanoblocks were both resolvable. For the analysis of the relationship between potential release sites, receptor-like particles, and PSD nanoblocks, it was conducted in the 13 synaptosomes, where PSDs were resolvable, potential release sites were detectable, and subtomogram averaging identified receptor-like particles exist. Nanoblocks were classified into four different types by calculating if a particle falls into the convex hull of a nanoblock: type B nanoblock, which only contains type B particles; type A nanoblock, which only contain type A particles; mixed nanoblock containing both type A and type B particles; and nanoblock without detected type A or type B particles. To test if the alignment between potential release sites and PSD nanoblock is specialized, the position of the vesicles was randomized within the bounds of the entire PSD. We randomized each synaptic vesicle position one hundred times and then performed the analysis for the randomized vesicle position to calculate $d1/D$ as we did for actual vesicles.

### Statistical analysis

Statistical analysis was performed as indicated in legends with OriginPro 2024. Comparisons of the relationship of normalized distance $d1/D$ between type A particles and nanoblocks and $d1/D$ between type B particles and nanoblocks were tested by two-sample Kolmogorov–Smirnov test (Fig. 4 D and in the text) and Mann–Whitney test (in the text). Comparisons of the relationship of the relationship of normalized distance $d1/D$ between type A and B particles and nanoblocks and $d1/D$ between undefined particles and nanoblocks were tested by two-sample Kolmogorov–Smirnov test (Fig. 4 F and in the text) and Mann–Whitney test (in the text). Comparisons of the relationship of normalized distance $d2/D$ between the experimental potential release sites and nanoblocks and $d2/D$ between randomized potential release site and nanoblock in intact synapses from cultured neurons were tested by two-sample Kolmogorov–Smirnov test (Fig. 5 C and in the text). Comparisons of the relationship of normalized distance $d2/D$ between the experimental potential release sites and nanoblocks and $d2/D$ between randomized potential release sites and nanoblocks in synaptosomes were tested by two-sample Kolmogorov–Smirnov test (Fig. 5 D and in the text). Comparisons of the relationship of normalized distance $d2/D$ between potential release sites and centroids of different types of nanoblocks were tested by Kruskal–Wallis ANOVA (Fig. 5 F and in the text). Comparisons of the relationship of areas of different types of nanoblocks were tested by Kruskal–Wallis ANOVA (Fig. 5 G and in the text). In the two-sample Kolmogorov–Smirnov test, D represents the maximum absolute difference between the cumulative distribution functions of the two samples being compared, indicating the extent of their distributional divergence.

### Online supplemental material

Fig. S1 shows the cryo-ET of distinct stages of vesicles in excitatory synapse, inhibitory synapse, synaptic protein–like particles, and PSD densities. Fig. S2 shows the IsoNet-corrected tomograms with distinct iterations and DBSCAN clustering results with distinct parameters. Fig. S3 shows the quantification of synaptic components in different sample preparations of synaptosomes. Video 1 shows the 3D surface rendering of a synapse in primary cultured neurons observed by cryo-ET. Video 2 shows the 3D surface rendering of a synaptosome observed by cryo-ET with type A and type B particles anchored on the postsynaptic membrane.

### Data availability

All EM data were deposited in EMPIAR-12482. Type A density map was deposited as EMD-48114, and type B density map was deposited as EMD-48113. Custom code for data analysis was uploaded on GitHub https://github.com/qiangjunzhou-cryoET/Postsynaptic_density-Tomo. The original and/or analyzed data sets generated during the current study are available from the corresponding author upon reasonable request.

## Acknowledgments

We thank Drs. Ege Kavalali, Terunaga Nakagawa, William Wan, Roger Colbran, and Mark Bowen for insightful discussions.

This work was supported by the National Institutes of Health (NIH) (R00 MH113764 and R01 MH132918 to Q. Zhou), the Vanderbilt Faculty Fellowship Endowment Fund, and the Vanderbilt Brain Institue "Neurodegenerative" TIPS Initiative Award for support. EM data collection was performed at the Vanderbilt Center for Structural Biology Cryo-EM Facility,

including the Glacios cryo-TEM funded by NIH grant S10 OD030292-01. Open Access funding provided by Vanderbilt University.

Author contributions: R. Sun: conceptualization, data curation, formal analysis, investigation, methodology, project administration, software, validation, visualization, and writing—original draft, review, and editing. J.P. Allen: data curation, formal analysis, and writing—review and editing. Z. Mao: data curation and software. L. Wilson: formal analysis, investigation, visualization, and writing—review and editing. M. Haider: conceptualization, formal analysis, investigation, and writing—review and editing. B. Alten: investigation and writing—review and editing. Z. Zhou: data curation, methodology, and writing—review and editing. X. Wang: investigation. Q. Zhou: conceptualization, data curation, formal analysis, funding acquisition, investigation, methodology, project administration, resources, supervision, validation, visualization, and writing—original draft,review, and editing.

Disclosures: The authors declare no competing interests exist.

Submitted: 23 June 2024

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

# Supplemental material

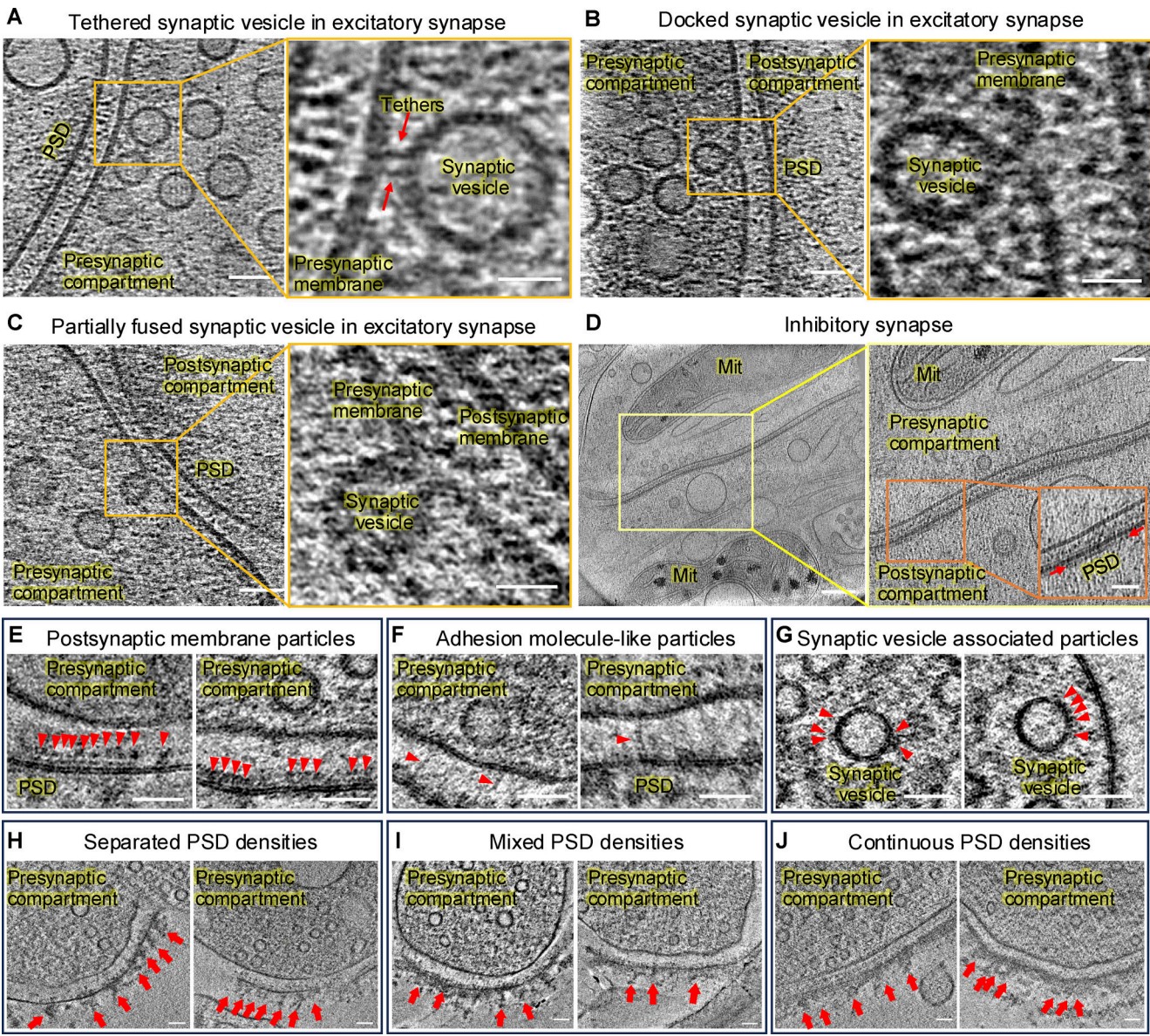

Figure S1. **Cryo-ET of distinct stages of vesicles in excitatory synapse, inhibitory synapse, synaptic protein–like particles, and PSD densities. (A–C)** Synaptic vesicles at different stages of exocytosis show potential release sites on the presynaptic membrane in excitatory synapses. **(D)** The tomogram slice shows a typical inhibitory synapse with thin PSD. Red Arrows in the inset indicate the PSD. **(E–G)** Tomographic slices show postsynaptic membrane particles (E), adhesion molecule–like particles (F), and synaptic vesicle–associated particles (G), as arrowheads indicate in synaptosomes. **(H)** Synaptosomes with separate PSD density clusters. **(I)** Synaptosomes with separate and continuous PSD densities. **(J)** Synaptosomes with continuous PSD densities. Arrows indicate PSD density clusters. Scale bars: (A–C) left panels, 50 nm; right panels, 20 nm; (D) left panel, 200 nm; right panel, 200 nm; inset, 50 nm; (E–G) 50 nm; (H–J) 100 nm.

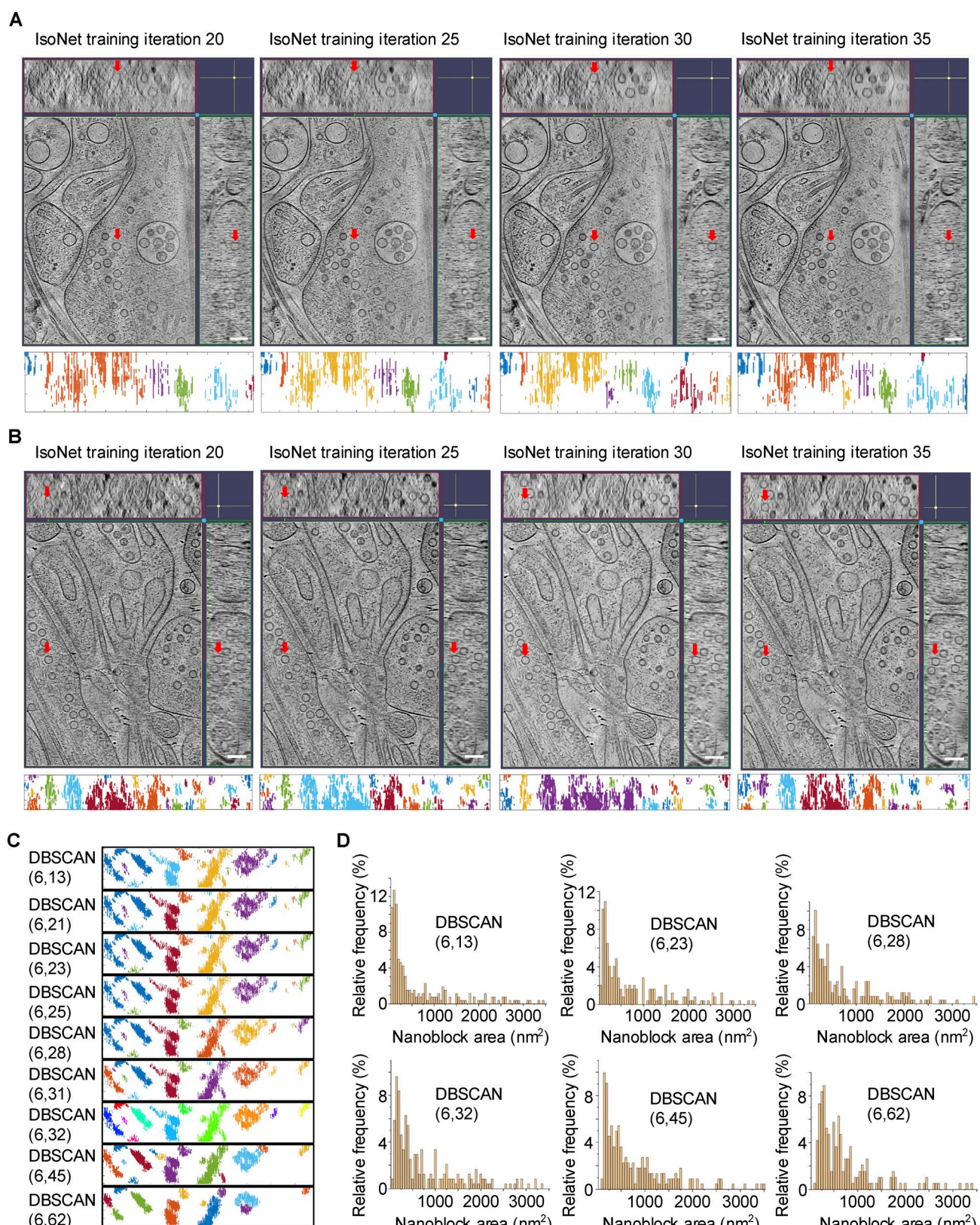

Figure S2. **IsoNet-corrected tomograms with distinct iterations and DBSCAN clustering results with distinct parameters. (A and B)** Two examples showing IsoNet-corrected tomograms with distinct iterations. Red arrows indicate the same vesicle in different planes. Scale bars: 100 nm. **(C)** DBSCAN clustering results of an excitatory PSD with different DBSCAN parameters. **(D)** Distribution of nanoblock area in synaptosome (without protease inhibitor or DTT) prep 1 with different DBSCAN parameters.

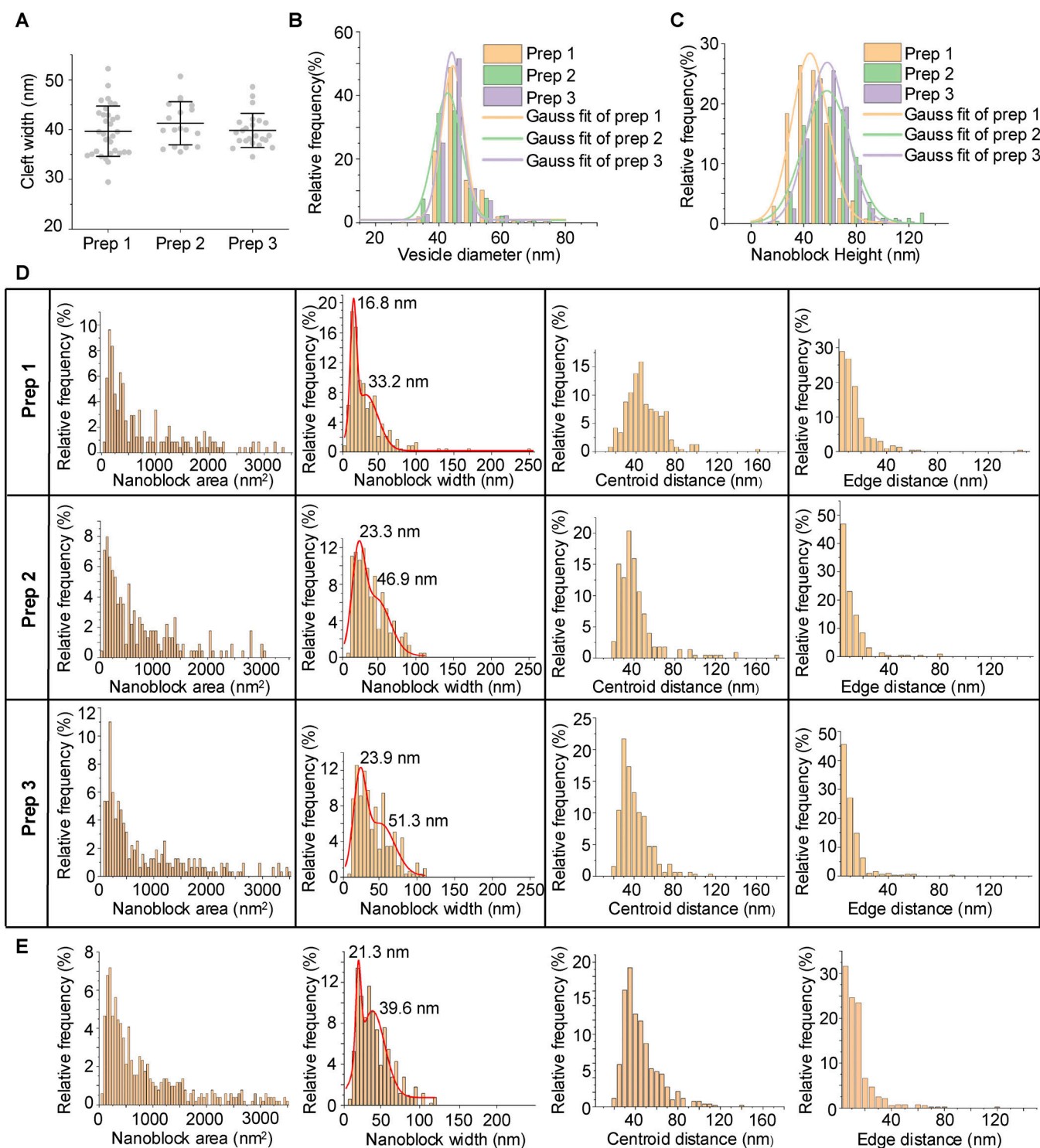

Figure S3. **Quantification of synaptic components in different sample preparations of synaptosomes. (A)** Cleft widths across three preparations ($n$ = 33 for prep 1, $n$ = 18 for prep 2, and $n$ = 23 for prep 3) without protease inhibitor or DTT with mean ± SD of each sample shown in the figure. **(B)** Synaptic vesicle size distribution for three preparations. **(C)** Nanoblock height distribution for three preparations. **(D)** Distribution of PSD nanoblock area, width, centroid-to-centroid distance, and edge-to-edge distance across three preparations. **(E)** Distribution of PSD nanoblock area, width, centroid-to-centroid distance, and edge-to-edge distance in synaptosomes isolated with protease inhibitor and DTT.

Video 1. **3D surface rendering of a synapse in primary cultured neurons observed by cryo-ET.** Spherical structures (cyan) are synaptic vesicles. Other subcellular structures are presynaptic membrane (green), postsynaptic membrane (deep blue), PSD (gold), microtubules (light blue), endoplasmic reticulum (purple), presynaptic endosomes (red), and postsynaptic endosome (red). Video frame rate: 18 fps. Scale bar: 200 nm.

Video 2.   **3D surface rendering of a synaptosome observed by cryo-ET with type A and type B particles anchored on the postsynaptic membrane.**
Spherical structures (cyan) are synaptic vesicles. Other subcellular structures are presynaptic membrane (green), postsynaptic membrane (deep blue), PSD (gold), type A particles (yellow), and type B particles (red). Video frame rate: 23.98 fps. Scale bar: 100 nm.

