## [Peer Review File · The Journal of Cell Biology]

The postsynaptic density in excitatory synapses is composed of clustered, heterogeneous nanoblocks

Rong Sun, James Allen, Zhuqing Mao, Liana Wilson, Mariam Haider, Baris Alten, Zimeng Zhou, Xinyi Wang, and Qiangjun Zhou

Corresponding Author(s): Qiangjun Zhou, Vanderbilt University

Review Timeline:

Submission Date:	2024-06-23
Editorial Decision:	2024-08-15
Revision Received:	2024-12-05
Editorial Decision:	2025-01-07
Revision Received:	2025-01-14

Monitoring Editor: Eva Nogales

Scientific Editor: Dan Simon

Transaction Report:

DOI: <https://doi.org/10.1083/jcb.202406133>

August 15, 2024

Re: JCB manuscript #202406133

Dr. Qiangjun Zhou
Vanderbilt University School of Medicine
Department of Cell and Developmental Biology
7158C MRB3
465 21st Avenue South
Nashville, TN 37232

Dear Dr. Zhou,

Thank you for submitting your manuscript entitled "Cryo-electron tomography reveals postsynaptic nanoblocks in excitatory synapses." The manuscript was assessed by expert reviewers, whose comments are appended to this letter. We invite you to submit a revision as a Report, if you can address the reviewers' key concerns, as outlined here.

You will see that the reviewers feel the study provides an important contribution to our understanding of the postsynaptic density organization. The major request is for a more comprehensive analysis of the composition of the synaptoneuroosomes to compare with synaptosomes with detached PSDs. Other comments ask for clarifications of the tomogram reconstruction and subtomogram averaging processes, statistical analysis, more methods details, additional comparison of your findings with prior studies, and other changes to text and figures.

GENERAL GUIDELINES:

Text limits: Character count for a Report is < 20,000, not including spaces. Count includes title page, abstract, introduction, the joint Results & Discussion, and acknowledgments. Count does not include materials and methods, figure legends, references, tables, or supplemental legends.

Figures: Reports may have up to 5 main text figures. To avoid delays in production, figures must be prepared according to the policies outlined in our Instructions to Authors, under Data Presentation, <https://jcb.rupress.org/site/misc/ifora.xhtml>. All figures in accepted manuscripts will be screened prior to publication.

Supplemental information: There are strict limits on the allowable amount of supplemental data. Reports may have up to 3 supplemental figures. Up to 10 supplemental videos or flash animations are allowed. A summary of all supplemental material should appear at the end of the Materials and methods section.

Please note that JCB now requires authors to submit Source Data used to generate figures containing gels and Western blots with all revised manuscripts. This Source Data consists of fully uncropped and unprocessed images for each gel/blot displayed in the main and supplemental figures. If your revised paper will include cropped gel and/or blot images, please be sure to provide one Source Data file for each figure that contains gels and/or blots along with your revised manuscript files. File names for Source Data figures should be alphanumeric without any spaces or special characters (i.e., SourceDataF#, where F# refers to the associated main figure number or SourceDataFS# for those associated with Supplementary figures). The lanes of the gels/blots should be labeled as they are in the associated figure, the place where cropping was applied should be marked (with a box), and molecular weight/size standards should be labeled wherever possible. Source Data files will be made available to reviewers during evaluation of revised manuscripts and, if your paper is eventually published in JCB, the files will be directly linked to specific figures in the published article.

The typical timeframe for revisions is three to four months. If you anticipate any difficulties in meeting this aforementioned revision time limit, please contact us and we can work with you to find an appropriate time frame for resubmission. Please note that papers are generally considered through only one revision cycle, so any revised manuscript will likely be either accepted or

rejected.

Thank you for this interesting contribution to Journal of Cell Biology. You can contact us at the journal office with any questions at cellbio@rockefeller.edu.

Sincerely,

Eva Nogales, PhD
Monitoring Editor
Journal of Cell Biology

Dan Simon, PhD
Scientific Editor
Journal of Cell Biology

Reviewer #1 (Comments to the Authors (Required)):

Sun and colleagues used cryo-electron tomography (Cryo-ET) to investigate the supramolecular organization of excitatory synapses. The authors examined the 3-dimensional architectures of postsynaptic densities (PSDs) and opposing nerve terminals in synapses of cultured neurons and biochemically isolated synaptosomes. They demonstrated that PSDs are composed of membrane-associated nanoblocks of various sizes. Furthermore, they showed that the sites of vesicular exocytosis are more likely to be positioned close to nanoblocks with specific receptor-like particles. This topic is important since the nanoscale architectures of neuronal synapses are yet to be fully resolved. The presented experiments are technically sound, and the results are mostly convincing. The paper would be a good candidate for JCB, provided the following points are appropriately addressed.

1. The idea of applying Cryo-ET to analyze semi-intact synapses isolated from the brain by subcellular fractionations is very exciting. However, the manuscript, as it stands, falls short of extensively benchmarking this preparation to exclude the possibility of artifacts. The authors present evidence that PSDs are composed of nanoblocks in both synaptosomes with detached postsynaptic compartments and more intact synaptoneuroosomes, but this evidence is somewhat scattered. Several concluding statements appear speculative since only analyses of synaptosomes are shown. The study would be much stronger if the compositions of presynaptic active zones, vesicles, clefts, PSDs, and other resolvable structures were systematically analyzed in all three preparations (cultured neurons and two types of biochemically isolated synapses) side-by-side.
2. Along the same lines, it would be useful to include more detailed analyses demonstrating the extent of variability (or lack thereof) in reconstructions from independent biological samples beyond what is shown in Figure S3.
3. Perhaps the most important conceptual implication of this study is that it reveals additional complexity in PSDs. While the authors' notion that the previously described nanodomains are further subdivided into nanoblocks is interesting, this argument is largely based on size differences (e.g., nanoblocks are smaller). Could these differences be attributed to variations in experimental procedures and/or imaging techniques used?
4. Line 163: Considering the relatively low resolution of reconstructions used in the study, it is a stretch to say that "specific synaptic proteins" were visualized.
5. Some panels in Figure 1 are redundant.
6. In general, the presentation of main and supplemental data is not helpful. The figures and legends are shown separately in different places, making it hard to read the manuscript.
7. The authors should edit the text to improve clarity and correct typos.

Reviewer #2 (Comments to the Authors (Required)):

In this manuscript, the authors utilized cryo-electron tomography to analyze the 3D structural features of the postsynaptic components in isolated synaptosomes and intact synapses from cultured neurons. The molecular architecture of postsynaptic density (PSD) is a complicated system for cryo-electron tomographic analysis. The authors projected a 40nm-thick PSD slab roughly parallel to the synaptic membrane for a 2D density map to be used for density domain (termed nanoblock in the manuscript) determination and analysis with the DBSCAN data clustering algorithm. This strategy of quantitative characterization represents a methodology novelty in this manuscript. Meanwhile, the authors employed sub-volume averaging to explore the possible molecular identities and distributions of membrane components, and they combined the results with the information from the PSD domain analysis to try to extract biological insight. Overall, the manuscript represents a significant effort in cryo-EM specimen preparation and cryo-tomographic analysis. The data quality is very good. Some of the analyses require further refinement and verification. Although the manuscript is written in a descriptive style with limited novel biological insight, the structural data of the tomograms, the domain (nano-block) analysis strategy, and the type A and type B sub-volumes obtained by the sub-tomogram averaging are valuable resources that would benefit further PSD studies, therefore, should be published. However, the authors should address the following issues before a publication:

1. As shown in Fig 1G, there is still some missing wedge effect after IsoNet reconstruction, which is anticipated. However, the variation in the degree of the effect will likely affect the nanoblock analysis results, making a quantitative comparison among different batches of samples/data sets (under different tilt-angle ranges for example) very difficult. The manuscript did not provide the necessary details for the cryo-tomogram reconstruction using IsoNet. The reconstruction outcomes using IsoNet for missing wedge compensation do depend on input parameters. For example, inappropriate iteration numbers sometimes result in noticeable artifacts, which may affect the domain analysis results using the DBSCAN algorithm. I wonder if the authors have tried comparing the DBSCAN results under different IsoNet reconstruction parameters for the same (and different) cryo-tomogram(s) to determine the evaluate the sensitivity to the IsoNet input parameters and see if it is necessary to introduce a step of IsoNet parameter optimization/standardization for consistent DBSCAN results.

2. The manuscript did not provide the necessary details for the application of the DBSCAN clustering work, although I considered it as a method novelty. The Methods section did not clarify how DBSCAN algorithm was "adopted to do the clustering of PSD density". What package and version did the authors choose to use (e.g., MATLAB, R, pyclustering...)? What parameters were used (eps and the minimum number of points?) Although the outcomes are relatively insensitive to the input parameters, these are the basic information to be included in the Methods section.

3. The authors manually selected 1,565 protein particles on the postsynaptic membrane from 28 synaptosomes with the sizes of extracellular parts ranging from 10 nm to 14 nm for subtomogram averaging. They achieved an averaged type A structure with 391 particles and an averaged type B structure with 189 particles. Therefore, 985 protein particles are excluded from the subtomogram averaging, which is reasonable. After the subtomogram averaging, the authors superimposed the Type A and Type B particles and the docked and tethered or fused vesicles with the PSD clustering results. Here, the 985 excluded protein particles are not included in the analysis. In Fig 4D, I noticed that most particles are outside the nanoblocks or at the edge of the nanoblock densities. Where are the excluded 985 protein particles? It is possible that the particles buried in a strong background, such as in the nanoblock density domains, are more likely to offer a poor matching score and be excluded during subtomogram averaging procedure. In fact, manual particle picking is likely to give a bias to the particles in the background-free regions because they are easier to identify visually. Therefore, Fig.1D may not represent a realistic distribution of the Type A or Type B particles in relation to the nanoblock maps. The statements relied on this distribution analysis become questionable. A different strategy, such as template-matching of the averaged subvolumes of Type A and Type B against the original tomogram, may be needed for this analysis.

4. While the authors are cautious in relating the nanoblocks to the nanocolumns or nanodomains reported previously by other groups, the major conclusion is that the PSD is composed of clusters/domains of various sizes, assemblies, and distributions. There is not much new information or concepts provided. The suggestion that potential presynaptic release sites are closer to nanoblocks with type A/B receptor-like 27 particles than those without type A/B receptor-like particles is a speculation based on the particle distribution analysis that is questionable. Therefore, the manuscript is relatively weak regarding the biological insight that originated from this work, despite high-quality data and significant effort in quantitative tomogram analysis. The authors may consider to compare and discuss the size, distribution, et al. of the "nanoblocks" described in this manuscript with the nanodomains or nanocolumns in previous publications, and hopefully, this would help relate the results in this work with the biological insights published by others in the discussion.

5. The author projected a 40nm-thick PSD slab along the y-axis, for a 2D density projection image parallel to the synaptic membrane (x,z plane) and used it for column density domain (nano-block) analysis with the DBSCAN. Figures 1D to 1H try to explain how this was done technically, but it is not straightforward for readers without much structural training to understand. It will be helpful to label the axes (x,y) at the low-left corner of Figs. 1D and 1E. The author may place Fig 1F beneath Fig 1E and illustrate it as the 2D to the 1D projection of a single slice of the blue rectangle region. To gain space for this modification, the Fig 1A experimental route may be modified from an "-" configuration to a "C" configuration by moving the images of Vitrobot and Krios below the petri dish and the rat pup, respectively. In fact, I would be better to remove both the images of Vitrobot and Krios as they offered very little helpful information and then place Fig 1B and Fig 1C below Fig 1A for a better workflow demonstration. The same idea can be applied to the Fig 2.

6. The results of the subtomogram averaging and the raw tomographic data should be deposited into the EM data bank (EMDB) and the Electron Microscopy Public Image Archive (EMPIAR), respectively. The access IDs should be included in the manuscript.

7. In the method session, the authors only described the usage of the Krios 4 electron microscope, but they acknowledged the Glacios in the acknowledgment. Please address this inconsistency.

RESPONSE TO COMMENTS

The editor and reviewer comments are presented in *black italic font*, while our responses are provided in blue font.

Reviewer #1:

Sun and colleagues used cryo-electron tomography (Cryo-ET) to investigate the supramolecular organization of excitatory synapses. The authors examined the 3-dimensional architectures of postsynaptic densities (PSDs) and opposing nerve terminals in synapses of cultured neurons and biochemically isolated synaptosomes. They demonstrated that PSDs are composed of membrane-associated nanoblocks of various sizes. Furthermore, they showed that the sites of vesicular exocytosis are more likely to be positioned close to nanoblocks with specific receptor-like particles. This topic is important since the nanoscale architectures of neuronal synapses are yet to be fully resolved. The presented experiments are technically sound, and the results are mostly convincing. The paper would be a good candidate for JCB, provided the following points are appropriately addressed.

Response: We thank the Reviewer for the positive comments.

Comment 1: *The idea of applying Cryo-ET to analyze semi-intact synapses isolated from the brain by subcellular fractionations is very exciting. However, the manuscript, as it stands, falls short of extensively benchmarking this preparation to exclude the possibility of artifacts. The authors present evidence that PSDs are composed of nanoblocks in both synaptosomes with detached postsynaptic compartments and more intact synaptoneuroosomes, but this evidence is somewhat scattered. Several concluding statements appear speculative since only analyses of synaptosomes are shown. The study would be much stronger if the compositions of presynaptic active zones, vesicles, clefts, PSDs, and other resolvable structures were systematically analyzed in all three preparations (cultured neurons and two types of biochemically isolated synapses) side-by-side.*

Response: We appreciate the reviewer's constructive suggestions. As suggested, we have quantitatively analyzed synaptic vesicle diameter, cleft width, PSD nanoblock area, nanoblock width, nanoblock centroid-to-centroid distance, and nanoblock edge-to-edge distance across all three preparations (synaptosome, synaptoneurosome, and intact synapse) (Fig. 1I-J, Fig. 2E). Additionally, we conducted additional analyses on synaptosomes isolated with protease inhibitors and DTT (Fig. S3E) and compared with synaptosome samples prepared without protease inhibitors and DTT, showing nanoblocks with similar patterns of dimensions (Fig. S3D, E). For compositions of presynaptic active zones, we focused on tethered, docked, and partially fused synaptic vesicles which likely indicate the potential location of synaptic release sites. We analyzed the distribution of potential release sites with nanoblocks (Fig. 5C-D) and with receptor-like particles and nanoblocks (Fig. 5F-G). These updates provide more robust evidence that PSDs are composed of nanoblocks both in isolated synaptic terminals and in intact synapses of cultured neurons.

Though we put PSD nanoblock analyses side by side, the absolute values of the nanoblock parameters across different preparations are not strictly comparable due to variations in data collection and analysis methods. Specifically, tilt series for synaptosomes and synaptoneuroosomes were acquired using a phase plate, while those for synapses were collected without one. Additionally, only the tomograms of synapses were processed with IsoNet, as their relatively low contrast without the phase plate necessitated this step. We also applied different clustering parameters for the analysis as stated in the method section (Lines 580-598): a 0.25 gray value threshold with DBSCAN parameters of (eps = 6, minimum points = 32) for synaptosomes isolated without protease inhibitor or DTT, and a 0.2 gray value threshold

with DBSCAN parameters of (eps = 7, minimum points = 32) for synaptosome isolated with protease inhibitor and DTT, synaptoneurosomes and synapses. These parameters were optimized to ensure visual consistency with our observations.

Overall, these analyses confirm that the segregation of PSD densities is not an artifact and further demonstrate that synaptosomes and synaptoneurosomes represent unique and relatively stable preparations derived from brain tissue.

Comment 2: *Along the same lines, it would be useful to include more detailed analyses demonstrating the extent of variability (or lack thereof) in reconstructions from independent biological samples beyond what is shown in Figure S3.*

Response: Thanks for the valuable suggestions. To comprehensively demonstrate the variability among different synaptosome preparations, we analyzed synaptosome samples prepared without protease inhibitors and DTT from three independent experiments (Fig. S3). The cleft width and synaptic vesicle diameter are shown in Fig. S3A and B, while nanoblock height, area, width, centroid-to-centroid distance, and edge-to-edge distance are presented in Fig. S3C and D. Additionally, we analyzed synaptosomes isolated using iso-osmotic homogenization buffer with protease inhibitors and DTT, comparing their nanoblock area, width, centroid-to-centroid distance, and edge-to-edge distance in Fig. S3E. These results demonstrate the consistency of PSD nanoblocks and collectively highlight their dimensions, as well as the consistency and variability across sample preparations.

Comment 3: *Perhaps the most important conceptual implication of this study is that it reveals additional complexity in PSDs. While the authors' notion that the previously described nanodomains are further subdivided into nanoblocks is interesting, this argument is largely based on size differences (e.g., nanoblocks are smaller). Could these differences be attributed to variations in experimental procedures and/or imaging techniques used?*

Response: Thanks for the reviewer's positive comment and insightful suggestion. In response, we have added a statement about the complexity of the PSD (Lines 109-110 and 381-383). As suggested by Reviewer #2, we used more precise terminology in the text to discuss the comparison between nanoblocks and nanodomains (Lines 370-394). The median size (~30 nm width along the x-axis) of nanoblocks is smaller than the nanodomains previously reported via fluorescence imaging. However, larger nanoblocks (~60 nm width along the x-axis, Fig. 2D,E) are comparable in size to nanodomains, which range from 70 to 160 nm (Broadhead et al., 2016; Hruska et al., 2018; MacGillavry et al., 2013; Nair et al., 2013; Tang et al., 2016). These numbers suggest that PSD nanoblocks could be a smaller subsynaptic composition of the PSD compared to nanodomains. However, we also consider the possibility that nanoblocks of different sizes may have distinct compositions and functional roles, as discussed in the text (Lines 387-390). Some larger nanoblocks are likely to be nanodomains, while smaller nanoblocks may not be part of nanodomains. Additionally, we discussed the variations in experimental procedures and imaging techniques in the discussion section (Lines 390-394). Since these data are derived from different experiments and imaging methods, it is challenging to make strict comparisons. Further exploration of the composition of PSD nanoblocks will help us uncover the distinctions and similarities among nanoblocks, nanodomains, nanomodules and PSD part of nanocolumn.

Comment 4: *Line 163: Considering the relatively low resolution of reconstructions used in the study, it is a stretch to say that "specific synaptic proteins" were visualized.*

Response: Thanks for the valuable suggestion. Now instead of using "specific synaptic proteins", we use "synaptic protein-like particles" (Line 635, Fig. S1 legend) and "postsynaptic

membrane particles”, “adhesion molecule-like particles/densities” and “synaptic vesicle associated particles” (Lines 1021-1027 and 639-640, Fig. 5 and S1 legends).

Comment 5: *Some panels in Figure 1 are redundant.*

Response: Thanks for the helpful comment. We have revised Fig. 1 and removed redundant panels to enhance clarity.

Comment 6: *In general, the presentation of main and supplemental data is not helpful. The figures and legends are shown separately in different places, making it hard to read the manuscript.*

Response: Thanks for the helpful comment. To improve readability, we have added figure legends below each figure and supplementary figure, and we have also included these legends in the text.

Comment 7: The authors should edit the text to improve clarity and correct typos.

Response: Thanks for the suggestion. We have revised the text to improve clarity and corrected the typos from the previous manuscript.

Reviewer #2:

In this manuscript, the authors utilized cryo-electron tomography to analyze the 3D structural features of the postsynaptic components in isolated synaptosomes and intact synapses from cultured neurons. The molecular architecture of postsynaptic density (PSD) is a complicated system for cryo-electron tomographic analysis. The authors projected a 40nm-thick PSD slab roughly parallel to the synaptic membrane for a 2D density map to be used for density domain (termed nanoblock in the manuscript) determination and analysis with the DBSCAN data clustering algorithm. This strategy of quantitative characterization represents a methodology novelty in this manuscript. Meanwhile, the authors employed sub-volume averaging to explore the possible molecular identities and distributions of membrane components, and they combined the results with the information from the PSD domain analysis to try to extract biological insight. Overall, the manuscript represents a significant effort in cryo-EM specimen preparation and cryo-tomographic analysis. The data quality is very good. Some of the analyses require further refinement and verification. Although the manuscript is written in a descriptive style with limited novel biological insight, the structural data of the tomograms, the domain (nano-block) analysis strategy, and the type A and type B sub-volumes obtained by the sub-tomogram averaging are valuable resources that would benefit further PSD studies, therefore, should be published.

Response: We thank the Reviewer for the positive comments.

Comment 1: *As shown in Fig 1G, there is still some missing wedge effect after IsoNet reconstruction, which is anticipated. However, the variation in the degree of the effect will likely affect the nanoblock analysis results, making a quantitative comparison among different batches of samples/data sets (under different tilt-angle ranges for example) very difficult. The manuscript did not provide the necessary details for the cryo-tomogram reconstruction using IsoNet (details of reconstruction). The reconstruction outcomes using IsoNet for missing wedge compensation do depend on input parameters. For example, inappropriate iteration numbers sometimes result in noticeable artifacts, which may affect the domain analysis results using the DBSCAN algorithm. I wonder if the authors have tried comparing the DBSCAN results under different*

IsoNet reconstruction parameters for the same (and different) cryo-tomogram(s) to determine the evaluate the sensitivity to the IsoNet input parameters and see if it is necessary to introduce a step of IsoNet parameter optimization/standardization for consistent DBSCAN results.

Response: Thanks for the thoughtful comments and suggestions. First, the tilt series of synapses were all acquired from -60 to 60 degree with 2-degree interval as stated in the method section (Lines 517-518). Therefore, for our dataset, the degree of the missing wedge effect should be consistent across different tomograms and different batches, both before and after IsoNet correction. Second, we have updated details of IsoNet correction in the Methods section of the manuscript (Lines 558-564): IsoNet correction was performed using binning 4 five-iteration SIRT reconstructed tomograms. CTF deconvolve was done using SnrFalloff 1.0 and DeconvStrength 1.0. Refinement was performed using CTF-deconvolved tomograms with 30 iterations and “noFilter” nose mode. The final predicted tomograms were generated using model_iter30.h5 as the trained model. We did not notice any artifact using these parameters. Following the reviewer's suggestion, we tested 20, 25, 30, and 35 iterations for refinement in IsoNet. As illustrated in Fig. S2A-B top panels, the correction effect becomes progressively stronger with increasing iterations. The synaptic vesicles appear more spherical from 20 to 25 to 30 iterations as red arrow indicate. Tomograms from the 30 and 35 iterations both demonstrated good quality in correcting the missing wedge effect. Therefore, the 30 iterations we used proved to be an optimal choice for our tomograms in IsoNet. Additionally, we performed DBSCAN analysis on tomograms from different iterations. The gray values and clustering results were found to be insensitive to the number of iterations (Fig. S2A-B, bottom panels). Based on all these results, we believe the strategy and parameters we used to collect tilt series and reconstruct tomograms with IMOD and IsoNet are optimal for DBSCAN analysis.

Comment 2: *The manuscript did not provide the necessary details for the application of the DBSCAN clustering work, although I considered it as a method novelty. The Methods section did not clarify how DBSCAN algorithm was “adopted to do the clustering of PSD density”. What package and version did the authors choose to use (e.g., MATLAB, R, pyclustering...)? What parameters were used (eps and the minimum number of points?) Although the outcomes are relatively insensitive to the input parameters, these are the basic information to be included in the Methods section.*

Response: Thanks for the insightful comments. MATLAB 2023b was used to apply internal DBSCAN algorithm for processing the projected gray values of the PSD. For synaptosomes isolated without protease inhibitor and DTT, DBSCAN (eps=6, minimum number of points =32) was applied. For synaptosome isolated with protease inhibitor and DTT, synaptoneurosome and synapse analysis, DBSCAN (eps=7, minimum number of points =32) was applied. We have updated the detailed information in the Methods section (Lines 580-598) and uploaded our custom MATLAB script to GitHub at https://github.com/qiangjunzhou-cryoET/Postsynaptic_density-Tomo.

Comment 3: *The authors manually selected 1,565 protein particles on the postsynaptic membrane from 28 synaptosomes with the sizes of extracellular parts ranging from 10 nm to 14 nm for subtomogram averaging. They achieved an averaged type A structure with 391 particles and an averaged type B structure with 189 particles. Therefore, 985 protein particles are excluded from the subtomogram averaging, which is reasonable. After the subtomogram averaging, the authors superimposed the Type A and Type B particles and the docked and tethered or fused vesicles with the PSD clustering results. Here, the 985 excluded protein particles are not included in the analysis. In Fig 4D, I noticed that most particles are outside the nanoblocks or at the edge of the nanoblock densities. Where are the excluded 985 protein particles? It is possible that the particles buried in a strong background, such as in the*

nanoblock density domains, are more likely to offer a poor matching score and be excluded during subtomogram averaging procedure. In fact, manual particle picking is likely to give a bias to the particles in the background-free regions because they are easier to identify visually. Therefore, Fig.1D may not represent a realistic distribution of the Type A or Type B particles in relation to the nanoblock maps. The statements relied on this distribution analysis become questionable. A different strategy, such as template-matching of the averaged subvolumes of Type A and Type B against the original tomogram, may be needed for this analysis.

Response: Thanks for the insightful comment and suggestion. We prepared a new panel Fig. 3B to better illustrate the subtomogram averaging process. Of the 985 excluded protein particles, 97 particles were excluded from undefined particles as they are either too close (<8nm) or bad particles that were not located on the postsynaptic membrane. For the remaining 888 undefined membrane particles, we analyzed 457 undefined particles across the 16 resolvable PSDs and updated the examples and statistics in Fig. 4A, E and F. These undefined particles are not significantly closer or farther from the nanoblock than type A&B particles (Mann-Whitney Test, $n=457$, median=0.84 for undefined particles and $n=300$, median=0.91 for type A and B particles, $p=0.1041$). In addition, 58.7% of type A&B particles were in the nanoblock range, while 60.0 % of undefined particles were in the nanoblock range (Fig. 4F). These results suggest manual picking and 3D classification together may have not introduced bias toward particles in the background-free regions.

Following the review's suggestion, we tried template matching using the software *pytom-match-pick* (Chaillat et al., 2024). We used low pass filtered O-shape AMPA receptor (PDB: 5IDE) with a membrane (Since the signal of the membrane was strong, we thought the density of the membrane may give a better matching in the beginning) for the template matching. However, the true positive is 0.5 based on *pytom-match-pick*'s internal criteria, even after instructing the software to pick 1200 particles in the example tomogram (red circles in Panel C) (see figure below). We also tried directly using the low pass filtered O-shape AMPA receptor (PDB: 5IDE) for template matching but this approach was unsuccessful (data not shown here). Due to the complexity of *in-situ* protein confirmation and the strong background noise in cellular tomography, we were unable to use template matching to identify AMPAR-like particles in our sample.

Currently, successful applications of template matching in cellular tomograms have primarily focused on protein supercomplexes, such as ribosomes and nuclear pore complexes (NPCs) (Cruz-León et al., 2024, Chaillet et al., 2023). These structures share a common feature: their substantial molecular weight, which exceeds several megadaltons. Their size and distinct structural features make them easier to identify within the dense and noisy environment of cells. In contrast, smaller or less rigid proteins present significant challenges for template matching in cellular tomography due to their lower signal-to-noise ratios and higher structural variability.

Comment 4: *While the authors are cautious in relating the nanoblocks to the nanocolumns or nanodomains reported previously by other groups, the major conclusion is that the PSD is composed of clusters/domains of various sizes, assemblies, and distributions. There is not much new information or concepts provided. The suggestion that potential presynaptic release sites are closer to nanoblocks with type A/B receptor-like particles than those without type A/B receptor-like particles is a speculation based on the particle distribution analysis that is questionable. Therefore, the manuscript is relatively weak regarding the biological insight that originated from this work, despite high-quality data and significant effort in quantitative tomogram analysis. The authors may consider to compare and discuss the size, distribution, et al. of the "nanoblocks" described in this manuscript with the nanodomains or nanocolumns in previous publications, and hopefully, this would help relate the results in this work with the biological insights published by others in the discussion.*

Response: Thanks for the insightful suggestion. We have added the comparison and included a discussion in the revised version. Specifically, on lines 370-394, we have included the discussion: "The development of imaging techniques such as super-resolution fluorescent microscopy has facilitated the discovery of subsynaptic protein clusters, e.g. *nanodomains* (Broadhead et al., 2016; Hruska et al., 2018; MacGillavry et al., 2013; Nair et al., 2013; Tang et al., 2016). These nanodomains range from 70 to 160 nm in size, larger than most PSD nanoblocks, which have a median width of ~30 nm within the PSDs of intact synapses and isolated synaptic terminals using cryo-ET (Fig. 2E). However, larger PSD nanoblocks (~60 nm widths along the x-axis, Fig. 2D,E) in intact synapses and isolated synaptic terminals (Fig. 2D, E) are comparable in size to PSD clusters in nanocolumns. In addition, larger nanoblocks are more effective at recruiting receptor-like particles and aligning with presynaptic release sites (Fig. 5). Thus, it is possible that larger nanoblock and nanodomain are the same structure observed through different imaging techniques. The heterogeneity in the size, assembly and spatial organization of PSD nanoblocks reveals further PSD complexity in response to activity-dependent molecular reorganization. A nanoblock may consist of certain copies of different scaffold proteins that form a supramolecular structure, linking to synaptic receptors and aligning with presynaptic release sites through adhesion molecules. Other postsynaptic proteins situated between these nanoblocks may play a role in regulating and organizing smaller nanoblocks into larger ones. The variation in the size and assembly of PSD nanoblocks might also reflect diverse protein compositions. It is also possible that nanoblocks of different sizes have distinct compositions and functional roles. Due to variations in experimental procedures and imaging techniques, and because these results are derived from different experiments, strict comparisons are challenging. Further exploration of the composition of PSD nanoblocks will help uncover the distinctions and similarities among nanoblocks, nanodomains, nanomodules, and the PSD portion of the nanocolumn."

Comment 5: *The author projected a 40nm-thick PSD slab along the y-axis, for a 2D density projection image parallel to the synaptic membrane (x,z plane) and used it for column density domain (nano-block) analysis with the DBSCAN. Figures 1D to 1H try to explain how this was done technically, but it is not straightforward for readers without much structural training to*

understand. It will be helpful to label the axes (x,y) at the low-left corner of Figs. 1D and 1E. The author may place Fig 1F beneath Fig 1E and illustrate it as the 2D to the 1D projection of a single slice of the blue rectangle region. To gain space for this modification, the Fig 1A experimental route may be modified from an "-" configuration to a "O" configuration by moving the images of Vitrobot and Krios below the petri dish and the rat pup, respectively. In fact, I would be better to remove both the images of Vitrobot and Krios as they offered very little helpful information and then place Fig 1B and Fig 1C below Fig 1A for a better workflow demonstration. The same idea can be applied to the Fig 2.

Response: Thanks for the suggestions. We have removed the images of Vitrobot and Krios and prepared new panels for current Fig. 1. We have also prepared a new Fig. 2A to include the coordinate system, which better demonstrates how we perform the analysis.

Comment 6: *The results of the subtomogram averaging and the raw tomographic data should be deposited into the EM data bank (EMDB) and the Electron Microscopy Public Image Archive (EMPIAR), respectively. The access IDs should be included in the manuscript.*

Response: Thanks for the valuable suggestions. We have deposited two subtomogram-averaged density maps in the EMD-48114 (type A receptor-like particle) and EMD-48113 (type-B receptor-like particle) and all images in EMPIAR. The corresponding access IDs have been provided in the manuscript.

Comment 7: *In the method session, the authors only described the usage of the Krios 4 electron microscope, but they acknowledged the Glacios in the acknowledgment. Please address this inconsistency.*

Response: Thanks for pointing out this issue. We used the Glacios microscope for sample screening and optimization in this study, and this detail has been included in the Methods section (Lines 527-528).

January 7, 2025

RE: JCB Manuscript #202406133R

Qiangjun Zhou
Vanderbilt University School of Medicine

Dear Dr. Zhou,

Thank you for submitting your revised manuscript entitled "Cryo-electron tomography reveals postsynaptic nanoblocks in excitatory synapses" which has now been reassessed by both of the original reviewers.

The reviewers consider that your manuscript is much improved and should be published in JCB. The second reviewer has a number of recommendations, all concerning style and wording, that we would like you to seriously consider to generate a final version for publication. Thus, we invite you to resubmit the manuscript either with the modifications as proposed or with arguments as to why you prefer to keep the manuscript as is. This should not take you very long and we will then be able to proceed to publication of your paper. Please also make any final changes that may be necessary to meet our formatting guidelines (see details below).

A. MANUSCRIPT ORGANIZATION AND FORMATTING:

1) Text limits: Character count for Reports is < 20,000, not including spaces. Count includes title page, abstract, introduction, combined results & discussion, and acknowledgments. Count does not include materials and methods, figure legends, references, tables, or supplemental legends.

**** Reports must have a single 'Results and Discussion' section. ****

2) Figure formatting: Reports may have up to 5 main text figures. Scale bars must be present on all microscopy images, including inset magnifications. Molecular weight or nucleic acid size markers must be included on all gel electrophoresis. Also, please avoid pairing red and green for images and graphs to ensure legibility for color-blind readers. If red and green are paired for images, please ensure that the particular red and green hues used in micrographs are distinctive with any of the colorblind types. If not, please modify colors accordingly or provide separate images of the individual channels.

3) Statistical analysis: Error bars on graphic representations of numerical data must be clearly described in the figure legend. The number of independent data points (n) represented in a graph must be indicated in the legend. Please, indicate whether 'n' refers to technical or biological replicates (i.e. number of analyzed cells, samples or animals, number of independent experiments). If independent experiments with multiple biological replicates have been performed, we recommend using distribution-reproducibility SuperPlots (please see Lord et al., JCB 2020) to better display the distribution of the entire dataset, and report statistics (such as means, error bars, and P values) that address the reproducibility of the findings.

Statistical methods should be explained in full in the materials and methods. For figures presenting pooled data the statistical measure should be defined in the figure legends. Please also be sure to indicate the statistical tests used in each of your experiments (both in the figure legend itself and in a separate methods section) as well as the parameters of the test (for example, if you ran a t-test, please indicate if it was one- or two-sided, etc.). Also, if you used parametric tests, please indicate if the data distribution was tested for normality (and if so, how). If not, you must state something to the effect that "Data distribution was assumed to be normal but this was not formally tested."

4) Title: While your current title may be appreciated by specialists, we do not feel that it will be accessible to a broader cell biology audience. To convey the advance more clearly, we suggest the following title: "The postsynaptic density in excitatory synapses is composed of clustered, heterogeneous nanoblocks"

5) Materials and methods: Should be comprehensive and not simply reference a previous publication for details on how an experiment was performed. Please provide full descriptions (at least in brief) in the text for readers who may not have access to referenced manuscripts. The text should not refer to methods "...as previously described."

6) For all cell lines, vectors, constructs/cDNAs, etc. - all genetic material: please include database / vendor ID (e.g. Addgene, ATCC, etc.) or if unavailable, please briefly describe their basic genetic features, even if described in other published work or

gifted to you by other investigators (and provide references where appropriate). Please be sure to provide the sequences for all of your oligos: primers, si/shRNA, RNAi, gRNAs, etc. in the materials and methods. You must also indicate in the methods the source, species, and catalog numbers/vendor identifiers (where appropriate) for all of your antibodies, including secondary. If antibodies are not commercial, please add a reference citation if possible.

7) Microscope image acquisition: The following information must be provided about the acquisition and processing of images:

- a. Make and model of microscope
- b. Type, magnification, and numerical aperture of the objective lenses
- c. Temperature
- d. Imaging medium
- e. Fluorochromes
- f. Camera make and model
- g. Acquisition software
- h. Any software used for image processing subsequent to data acquisition. Please include details and types of operations involved (e.g., type of deconvolution, 3D reconstitutions, surface or volume rendering, gamma adjustments, etc.).

8) References: There is no limit to the number of references cited in a manuscript. References should be cited parenthetically in the text by author and year of publication. Abbreviate the names of journals according to PubMed.

9) Supplemental materials: Reports may have up to 3 supplemental figures and 10 videos.

Please also note that tables, like figures, should be provided as individual, editable files. A summary of all supplemental material should appear at the end of the Materials and methods section. Please include one brief sentence per item.

10) Video legends: Should describe what is being shown, the cell type or tissue being viewed (including relevant cell treatments, concentration and duration, or transfection), the imaging method (e.g., time-lapse epifluorescence microscopy), what each color represents, how often frames were collected, the frames/second display rate, and the number of any figure that has related video stills or images.

11) eTOC summary: A ~40-50 word summary that describes the context and significance of the findings for a general readership should be included on the title page. The statement should be written in the present tense and refer to the work in the third person. It should begin with "First author name(s) et al..." to match our preferred style.

13) A separate author contribution section is required following the Acknowledgments in all research manuscripts. All authors should be mentioned and designated by their first and middle initials and full surnames. We encourage use of the CRediT nomenclature (<https://casrai.org/credit/>).

14) ORCID IDs: ORCID IDs are unique identifiers allowing researchers to create a record of their various scholarly contributions in a single place. Please note that ORCID IDs are required for all authors. At resubmission of your final files, please be sure to provide your ORCID ID and those of all co-authors.

15) Journal of Cell Biology now requires a data availability statement for all research article submissions. These statements will be published in the article directly above the Acknowledgments. The statement should address all data underlying the research presented in the manuscript. Please visit the JCB instructions for authors for guidelines and examples of statements at (<https://rupress.org/jcb/pages/editorial-policies#data-availability-statement>).

B. FINAL FILES:

-- A cover letter explaining how the remaining reviewer comments were addressed.

-- Cover images: If you have any striking images related to this story, we would be happy to consider them for inclusion on the journal cover. Submitted images may also be chosen for highlighting on the journal table of contents or JCB homepage carousel.

Images should be uploaded as TIFF or EPS files and must be at least 300 dpi resolution.

****It is JCB policy that if requested, original data images must be made available to the editors. Failure to provide original images upon request will result in unavoidable delays in publication. Please ensure that you have access to all original data images prior to final submission.****

****The license to publish form must be signed before your manuscript can be sent to production. A link to the electronic license to publish form will be sent to the corresponding author only. Please take a moment to check your funder requirements before choosing the appropriate license.****

Thank you for your attention to these final processing requirements. Please contact the journal office with any questions at cellbio@rockefeller.edu.

Thank you for this interesting contribution, we look forward to publishing your paper in Journal of Cell Biology.

Sincerely,

Eva Nogales, PhD
Monitoring Editor
Journal of Cell Biology

Dan Simon, PhD
Scientific Editor
Journal of Cell Biology

Reviewer #1 (Comments to the Authors (Required)):

The authors have done an excellent job addressing my concerns, making an already interesting paper even stronger. Their nanoscale analyses provide important insights into the organization of central synapses. I recommend this paper for publication.

Reviewer #2 (Comments to the Authors (Required)):

In this revised manuscript, the authors added technical details missing from the original version, and they invested effort to validate some of the analysis, modified figures, and provided more supplemental data. These efforts have addressed my previous concerns on the technical aspect of the analysis in the manuscript. In addition, the authors compared and related the "nanoblocks" in this manuscript with the "nanodomains" reported by others in the discussion. I think the revised manuscript has been improved and I would like to recommend it for publication.

The following are minor issues for the author to consider when polishing the final version.

There is still a lot of room to improve the Introduction part of the manuscript to help highlight the biological contribution, which is still relatively weak for this subcellular structural report. Additional background information should help readers, who are not familiar with the subcellular structures of synapses, appreciate the contribution of this work. For example, the current Introduction does not provide much biological background information about synapses (excitatory or inhibitory, and this manuscript focuses on excitatory synapses). What are the Type A/B receptor-like particles, and why does their distribution matter (related to a conclusion in this manuscript)? Suppose the authors want to emphasize the cryo-ET method application as it is in the manuscript now. In that case, the current status of cryo-ET applications in the subcellular structural studies of neuron cells and synapses should be discussed. These will help increase future citations of the work.

P2, line 38: "heterogeneous in size, assembly and distribution" change to "heterogeneous in size, assembly, and distribution"

P2, line 47: "... which likely underlies the dynamic nature of PSD to modulate synaptic strength." Consider rewriting! May change to "which likely underlies the dynamic nature of PSD"?

P2, line 52: "Typical examples of such nanoscale structures are neuronal chemical synapses". Consider: "One such example is

on neuronal chemical synapses"

P2, line 77": "However, nanoscale organization of the PSD and postsynaptic receptors is largely unknown....". Consider: "However, the precise nanoscale organization of the PSD and postsynaptic receptors is largely unknown...."

P3, line 82 to line 103: "To prevent introducing artifacts through chemical fixation, dehydration, and heavy-metal staining, biological samples are rapidly frozen to obtain a "true, instantaneous snapshot" (Bleck et al., 2010; Dubochet et al., 1988; Dubochet and Sartori Blanc, 2001; Taylor and Glaeser, 1974). In general, obtaining high-quality data requires cryo-ET samples to be thin enough for electrons to pass through, posing a technical challenge during sample preparation, which limits the resolution of cellular cryo-ET (Peet et al., 2019). To overcome these limitations, isolated synaptic terminals have been used for cryo-ET studies to decrease sample thickness (Fernandez-Busnadiego et al., 2010; Martinez-Sanchez et al., 2021). A classic type of isolated synaptic terminal called a synaptosome is highly suitable for cryo-ET, because it retains a piece of attached postsynaptic membrane that faces the active zone (Evans, 2015; Gray and Whittaker, 1962)." Please consider rewriting the entire portion. When rewrite, you may want to change the "electrons to pass through" to "electron beam to pass through" or "accelerated electrons to penetrate".

The sentences in line 86 to line 103 explain why synaptosomes were used for these studies besides the intact hippocampal neurons (both are used in this study). However, the current version is confusing with unclear statements and a jumpy logic link that should be revised. It can be difficult for general readers to understand what the "limitations" refer to in relation to the sentence "to overcome these limitations...": the challenge in preparing thin specimens for cryo-ET (while you can do cryo-ET with synapses in cultured hippocampal neurons)? the cultured cell specimens are not thin enough for a good signal-to-noise ratio? If the high thickness of the cells is the issue, why not use cryo-FIB (a known technique for people in the cryo-EM field)? If cutting thin by cryo-FIB is technically challenging and synaptosome isolation is relatively more reproducible and may minimize molecular crowding, thus was chosen as an alternative approach for this study, then clearly state it... Most readers will not be able to guess the technical information that is not clearly provided.

P4, line 133: "Therefore, after IsoNet correction, the PSD in Fig. 1C clearly..." should be "After IsoNet correction, the PSD in Fig. 1C clearly..."

P4, line 141: "After dissecting the hippocampi from rat brains, we performed four strokes of homogenization, followed by centrifugation at 800 g for 10 minutes (Fig. 1E). The resulting supernatant was plunge-frozen and used for cryo-ET data acquisition. Our preparations yielded two different types of isolated synaptic terminals based on their morphology: synaptoneuroosomes, which retained an enclosed postsynaptic compartment (Fig. 1E) (Hollingsworth et al., 1985; Quinlan et al., 1999), and synaptosomes, which retained a patch of postsynaptic membrane with PSD (Fig. 1E, see also Movie S2). In total, we acquired 123 tomograms of isolated synaptic terminals from three preparations in iso-osmotic homogenization buffer without protease inhibitor and DTT (99 synaptosomes and 24 synaptoneuroosomes). Their reduced sample thickness and lower molecular crowding provided higher-resolution structural information compared to intact synapses. This allowed us to directly visualize characteristic features of a synapse, including postsynaptic membrane particles, adhesion molecule-like particles and synaptic vesicle associated particles (Fig. S1E-G)." A large portion here are descriptions of methods instead of results and should be moved to Materials and methods.

P5, Line 184: "Visual inspection revealed three types of synaptosomes based on PSD organization: synaptosomes with separated PSD density clusters; synaptosomes with both separated and continuous PSD densities; and synaptosomes with continuous PSD densities (Fig.S1H-J)." Change to: "Visual inspection revealed three types of synaptosomes based on PSD organization: synaptosomes with separated PSD density clusters, synaptosomes with both separated and continuous PSD densities, and synaptosomes with continuous PSD densities (Fig.S1H-J)."

P9, line 350: "which we refer as PSD nanoblock" change to "which we refer to as a PSD nanoblock".

P26, the legend of Fig 1: Add in the description of Pannel (C) what the blue region refers to. (F) "A tomographic slice showing...". Uncorrected without IsoNet application? If so, clarify it in the legend.

RESPONSE TO COMMENTS

The editor and reviewer comments are presented in *black italic font*, while our responses are provided in blue font.

Reviewer #2 (Comments to the Authors (Required)):

In this revised manuscript, the authors added technical details missing from the original version, and they invested effort to validate some of the analysis, modified figures, and provided more supplemental data. These efforts have addressed my previous concerns on the technical aspect of the analysis in the manuscript. In addition, the authors compared and related the "nanoblocks" in this manuscript with the "nanodomains" reported by others in the discussion. I think the revised manuscript has been improved and I would like to recommend it for publication.

Response: We thank the Reviewer for the positive comments.

The following are minor issues for the author to consider when polishing the final version. There is still a lot of room to improve the Introduction part of the manuscript to help highlight the biological contribution, which is still relatively weak for this subcellular structural report. Additional background information should help readers, who are not familiar with the subcellular structures of synapses, appreciate the contribution of this work. For example, the current Introduction does not provide much biological background information about synapses (excitatory or inhibitory, and this manuscript focuses on excitatory synapses). What are the Type A/B receptor-like particles, and why does their distribution matter (related to a conclusion in this manuscript)? Suppose the authors want to emphasize the cryo-ET method application as it is in the manuscript now. In that case, the current status of cryo-ET applications in the subcellular structural studies of neuron cells and synapses should be discussed. These will help increase future citations of the work.

Response: Regarding the protein composition of type A/B receptor-like particles, we chose not to add further discussion to avoid overinterpreting our results. Currently, we discuss that type A and B receptor-like particles resemble AMPA receptors (Lines 245-247) and that their distributions align with previous studies on AMPAR distribution (Lines 337-343). However, these type A/B receptor-like particles were detected using subtomogram averaging and did not achieve sufficiently high resolution to fully confirm their protein compositions. To ensure careful interpretation of the results, we believe it is best to retain the current level of discussion.

Due to character limits, we did not include additional discussion on the current status of cryo-ET applications in subcellular structural studies of neuronal cells and synapses.

P2, line 38: "heterogeneous in size, assembly and distribution" change to "heterogeneous in size, assembly, and distribution"

Response: Thanks for the suggestion. We have revised the text as reviewer suggested (Line 32).

P2, line 47: "... which likely underlies the dynamic nature of PSD to modulate synaptic strength." Consider rewriting! May change to "which likely underlies the dynamic nature of PSD"?

Response: Thanks for the suggestion. We have revised the text as “which likely contribute to the dynamic nature of PSD in modulating synaptic strength” (Lines 32-33).

P2, line 52: "Typical examples of such nanoscale structures are neuronal chemical synapses". Consider: "One such example is on neuronal chemical synapses"

Response: Thanks for the suggestion. We have revised the text as reviewer suggested (Line 38).

P2, line 77": "However, nanoscale organization of the PSD and postsynaptic receptors is largely unknown....". Consider: " However, the precise nanoscale organization of the PSD and postsynaptic receptors is largely unknown...."

Response: Thanks for the suggestion. We have revised the text as reviewer suggested (Lines 76-77).

P3, line 82 to line 103: "To prevent introducing artifacts through chemical fixation, dehydration, and heavy-metal staining, biological samples are rapidly frozen to obtain a "true, instantaneous snapshot" (Bleck et al., 2010; Dubochet et al., 1988; Dubochet and Sartori Blanc, 2001; Taylor and Glaeser, 1974). In general, obtaining high-quality data requires cryo-ET samples to be thin enough for electrons to pass through, posing a technical challenge during sample preparation, which limits the resolution of cellular cryo-ET (Peet et al., 2019). To overcome these limitations, isolated synaptic terminals have been used for cryo-ET studies to decrease sample thickness (Fernandez-Busnadiego et al., 2010; Martinez-Sanchez et al., 2021). A classic type of isolated synaptic terminal called a synaptosome is highly suitable for cryo-ET, because it retains a piece of attached postsynaptic membrane that faces the active zone (Evans, 2015; Gray and Whittaker, 1962)." Please consider rewriting the entire portion. When rewrite, you may want to change the "electrons to pass through" to "electron beam to pass through" or "accelerated electrons to penetrate".

The sentences in line 86 to line 103 explain why synaptosomes were used for these studies besides the intact hippocampal neurons (both are used in this study). However, the current version is confusing with unclear statements and a jumpy logic link that should be revised. It can be difficult for general readers to understand what the "limitations" refer to in relation to the sentence "to overcome these limitations...": the challenge in preparing thin specimens for cryo-ET (while you can do cryo-ET with synapses in cultured hippocampal neurons)? the cultured cell specimens are not thin enough for a good signal-to-noise ratio? If the high thickness of the cells is the issue, why not use cryo-FIB (a known technique for people in the cryo-EM field)? If cutting thin by cryo-FIB is technically challenging and synaptosome isolation is relatively more reproducible and may minimize molecular crowding, thus was chosen as an alternative approach for this study, then clearly state it... Most readers will not be able to guess the technical information that is not clearly provided.

Response: Thanks for the suggestion. However, due to the character count limit for a JCB report and to avoid confusing readers, we have simplified this section in the introduction part. (Lines 79-90).

P4, line 133: "Therefore, after IsoNet correction, the PSD in Fig. 1C clearly..." should be "After IsoNet correction, the PSD in Fig. 1C clearly..."

Response: Thanks for the suggestion. We have revised the text as reviewer suggested (Line 114).

P4, line 141: "After dissecting the hippocampi from rat brains, we performed four strokes of homogenization, followed by centrifugation at 800 g for 10 minutes (Fig. 1E). The resulting supernatant was plunge-frozen and used for cryo-ET data acquisition. Our preparations yielded two different types of isolated synaptic terminals based on their morphology: synaptoneurosomes, which retained an enclosed postsynaptic compartment (Fig. 1E) (Hollingsworth et al., 1985; Quinlan et al., 1999), and synaptosomes, which retained a patch of postsynaptic membrane with PSD (Fig. 1E, see also Movie S2). In total, we acquired 123 tomograms of isolated synaptic terminals from three preparations in iso-osmotic homogenization buffer without protease inhibitor and DTT (99 synaptosomes and 24 synaptoneurosomes). Their reduced sample thickness and lower molecular crowding provided higher-resolution structural information compared to intact synapses. This allowed us to directly visualize characteristic features of a synapse, including postsynaptic membrane particles, adhesion molecule-like particles and synaptic vesicle associated particles (Fig. S1E-G)." A large portion here are descriptions of methods instead of results and should be moved to Materials and methods.

Response: Thanks for the suggestion. We have revised the text and moved the detailed descriptions to the *Materials and Methods* section (synaptic terminal isolation, Lines 384-397; number of tilt series collected, Lines 455-457), as recommended by the reviewer.

P5, Line 184: "Visual inspection revealed three types of synaptosomes based on PSD organization: synaptosomes with separated PSD density clusters; synaptosomes with both separated and continuous PSD densities; and synaptosomes with continuous PSD densities (Fig.S1H-J)." Change to: "Visual inspection revealed three types of synaptosomes based on PSD organization: synaptosomes with separated PSD density clusters, synaptosomes with both separated and continuous PSD densities, and synaptosomes with continuous PSD densities (Fig.S1H-J)."

Response: Thanks for the suggestion. We have revised the text as reviewer suggested (Lines 160-163).

P9, line 350: "which we refer as PSD nanoblock" change to "which we refer to as a PSD nanoblock".

Response: We have removed this sentence to eliminate characters as a JCB report.

P26, the legend of Fig 1: Add in the description of Pannel (C) what the blue region refers to. (F) "A tomographic slice showing...": Uncorrected without IsoNet application? If so, clarify it in the legend.

Response: Thanks for the suggestion. We have added the description of blue region in the legend of Fig 1C. We have also added "without IsoNet correction" in the legend of Fig 1F.